# Beyond the Final Layer: Attentive Multi-Layer Fusion for Vision Transformers

## Abstract

With the rise of large-scale foundation models, efficiently adapting them to downstream tasks remains a central challenge. Linear probing, which freezes the backbone and trains a lightweight head, is computationally efficient but often restricted to last-layer representations. We show that task-relevant information is distributed across the network hierarchy rather than solely encoded in any of the last layers. To leverage this distribution of information, we apply an attentive probing mechanism that dynamically fuses representations from all layers of a Vision Transformer. This mechanism learns to identify the most relevant layers for a target task and combines low-level structural cues with high-level semantic abstractions. Across 20 diverse datasets and multiple pretrained foundation models, our method achieves consistent, substantial gains over standard linear probes. Attention heatmaps further reveal that tasks different from the pre-training domain benefit most from intermediate representations. Overall, our findings underscore the value of intermediate layer information and demonstrate a principled, task-aware approach for unlocking their potential in probing-based adaptation.

## 1 Introduction

Foundation models have transformed machine learning across various domains, ranging from language (Devlin et al., 2019; Brown et al., 2020) to vision (Radford et al., 2021; Oquab et al., 2024), by providing powerful pretrained backbones trained on large, general-purpose datasets. How to adapt these models to specific downstream tasks most effectively remains a central question. Although *fine-tuning* and its parameter-efficient variants (e.g., LORA; Hu et al., 2022; Jia et al., 2022; Chen et al., 2022) have been proven to yield strong performance, these methods are computationally expensive and require adapting the weights of the backbone during training which changes the underlying model from general-purpose to task-specific. A lighter alternative is (linear) *probing* (Razavian et al., 2014; Yosinski et al., 2014; Kornblith et al., 2019b), where the backbone remains unchanged and a small head is trained on top of it. Probing is attractive in settings with limited resources, although its accuracy is typically inferior to fine-tuning (Kornblith et al., 2019b).

The standard linear probing approach operates exclusively on the final-layer representation, which in Vision Transformers (ViTs; Dosovitskiy et al., 2021) is typically represented by the `CLS` token. This design implicitly assumes that the `CLS` token encodes all task-relevant information. However, recent work challenges this assumption: Chen et al. (2024) show that attentive probing over final-layer patch tokens outperforms `CLS`-only approaches by facilitating task-dependent spatial information fusion. Similarly, DINOv2 (Oquab et al., 2024) demonstrates that concatenating `CLS` tokens from several of the last layers can surpass single-layer methods by exploiting some hierarchical information fusion. Together, these results suggest that information crucial for downstream tasks is distributed across layers and tokens rather than exclusively captured by the final `CLS` token representation.

If dependence on the final layer is a limiting factor, a potential solution is to fuse the information distributed across the different levels of model layers. ViTs process information across multiple layers: early layers capture low-level visual patterns and structural cues (e.g., edges, textures), whereas later layers encode high-level semantic concepts aligned with the pre-training objective (Raghu et al., 2021; Dorszewski et al., 2025). When downstream tasks differ from the pre-training domain, the

final layer likely discards structural or textural information that remains crucial for the target application, yet this information often persists in intermediate layers.

Recent work has begun to exploit intermediate representations for transfer learning (Tu et al., 2023; Wu et al., 2024) and parameter-efficient adaptation (Evci et al., 2022). Despite these advances, most approaches rely on simple aggregation strategies, such as concatenation, that produce overly large feature vectors or fail to adapt to task-specific requirements. Furthermore, the relevance of different layers varies substantially across task settings. While some more specialized domains, such as satellite imagery or medical images, benefit from low-level structural cues encoded in early or intermediate layers, others that include a broad set of natural images (e.g., CIFAR-100) require high-level semantic abstraction encoded in last layers. This variability underscores the need for more flexible mechanisms that effectively harness intermediate features in probing settings.

**Attentive Probe CLS + AP tokens of $\mathcal{L}$**

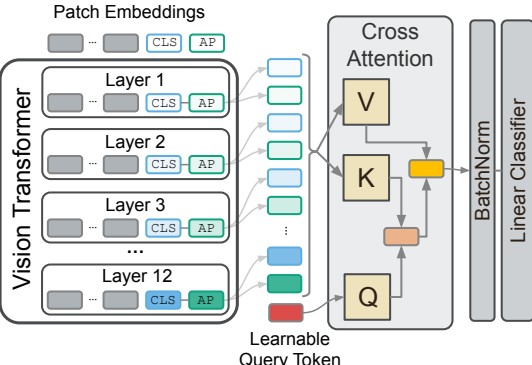

Figure 1: Schematic of our multi-layer Attentive Probe. The method applies cross-attention to `CLS` and `AP` tokens from multiple transformer layers, automatically discovering which representations contain the most task-relevant features.

In this work, we demonstrate how to effectively leverage valuable task-relevant information from a model's intermediate layers to substantially improve performance on diverse downstream tasks. Our evaluation shows that although intermediate layers hold valuable, complementary information, applying standard linear probing to representations from numerous layers leads to instability. This indicates a challenge in effectively combining features from a wide range of depths using a simple linear classifier. To solve this, we use an attention-based fusion method that dynamically weights the most informative layers for each task. Our approach considers both `CLS` and average-pooled (`AP`) tokens, combining semantic and aggregated spatial information. This method improves performance across various domains and, through the analysis of attention heatmaps, additionally helps us understand how different tasks use the model's hierarchical structure. Through evaluation across 20 diverse datasets and vision models from 3 different families, we find that different tasks adaptively leverage distinct layers of the representational hierarchy. Hierarchical fusion is particularly effective for datasets outside the pretraining domain and for models that compress information into summary tokens. In contrast, tasks that depend on localized spatial cues benefit from augmenting our approach with complementary spatial aggregation, highlighting the orthogonality between hierarchical and spatial information fusion. Fig. 1 provides an overview of our proposed multi-layer Attentive Probe.

In summary, our work makes three key contributions:

- We propose attentive probing using `CLS` and `AP` tokens from all intermediate layers, achieving consistent gains across 20 datasets with an average accuracy improvement of 5.54 percentage points compared to standard linear probing.

- We show that intermediate layer fusion provides consistent improvements across small, base, and large models, indicating that the approach generalizes across model scales without diminishing returns.

- We find that performance gains are largest for tasks that are different from the pre-training domain. Interpretable attention patterns show that natural image tasks rely more on later layers, while structural or specialized datasets benefit from intermediate representations, particularly from the `AP` tokens, underscoring the adaptive behavior of our probe.

## 2 RELATED WORK

ViTs (Dosovitskiy et al., 2021) pretrained on large-scale datasets, such as CLIP (Radford et al., 2021) and DINOv2 (Oquab et al., 2024), have become fundamental to computer vision. A key challenge is to efficiently adapt these foundation models to downstream tasks.

### 2.1 PROBING AND LIGHTWEIGHT ADAPTATION

Parameter-efficient fine-tuning (PEFT) aims to adapt large neural networks without updating all weights of the backbone. Popular methods include adapters (Chen et al., 2022), visual prompt tuning (Jia et al., 2022), and LoRA (Hu et al., 2022). An even more efficient paradigm is probing, where the entire pretrained backbone remains frozen, and only a lightweight module is trained on top of its features (Alain & Bengio, 2017). While early work focused on simple linear classifiers, recent studies have introduced more powerful attentive probes (Bardes et al., 2024; El-Nouby et al., 2024). This idea builds on earlier attention pooling methods such as the Set Transformer (Lee et al., 2019) and uses learnable attention modules to aggregate token features from the final layer (Yu et al., 2022; Chen et al., 2024; Psomas et al., 2025). However, their focus is confined to the final layer's output, implicitly assuming that the final representation is optimal for any given downstream task.

### 2.2 THE VALUE OF INTERMEDIATE REPRESENTATIONS

The principle that hierarchical features are crucial for robust recognition is fundamental to deep learning. In CNNs, representations progress from low-level patterns in early layers to high-level semantics in later ones (Zeiler & Fergus, 2014). The transferability differs across depth, with earlier layers being more general and later ones being more specialized (Yosinski et al., 2014). This led to iconic architectures, such as U-Net (Ronneberger et al., 2015) and Feature Pyramid Networks (Lin et al., 2017), which explicitly fuse features from shallow and deep layers to combine fine-grained details with high-level semantics. This principle extends to ViTs.

Although Raghu et al. (2021) showed that their representations are more uniform across layers than in CNNs and representation similarity evolves smoothly over depth (Lange et al., 2022), research confirms that ViT layers still gradually encode more complex semantic concepts (cf., Ghiasi et al., 2022; Dorszewski et al., 2025). Recognizing this, architectural extensions such as MViT (Fan et al., 2021), CrossViT (Chen et al., 2021), and Swin Transformer (Liu et al., 2021) explicitly integrate information at different resolutions. More recently, lightweight methods such as Head2Toe (Evci et al., 2022) and Visual Query Tuning (Tu et al., 2023) have shown that explicitly exploiting intermediate ViT layers can enhance transfer performance. Similar findings have emerged in language models, where probing has revealed that intermediate layers can even outperform the final layer depending on the task (Liu et al., 2019; Skean et al., 2025).

Building on these insights, we propose an adaptive attentive probe that learns to dynamically fuse representations from across the entire network hierarchy. Unlike prior work relying on fixed fusion schemes or manually chosen layer subsets, our method automatically discovers and weights the most task-relevant layers, combining the efficiency of probing with the power of adaptive multi-scale feature fusion.

## 3 METHOD

ViTs learn hierarchical feature representations where early layers capture low-level visual patterns while deeper layers encode high-level semantic concepts (Raghu et al., 2021). Conventional probing methods, which primarily use a model's last layers, may not be optimal for diverse downstream tasks because valuable, task-specific information often resides in intermediate layers. This mismatch necessitates adaptive layer selection to better align the model's feature representations with the specific requirements of the downstream task. We propose an attention-based fusion mechanism that dynamically weights and combines contributions from different transformer layers, allowing the model to automatically find the most relevant features for each downstream task.

**Problem Statement.** Consider a ViT encoder with $L$ attention layers processing an input image $x \in \mathbb{R}^{H \times W \times C}$. For each encoder layer $\ell \in \{1, \ldots, L\}$, we extract token embeddings

$\boldsymbol{Z}^{(\ell)} = [\boldsymbol{z}_0^{(\ell)}, \boldsymbol{z}_1^{(\ell)}, \ldots, \boldsymbol{z}_P^{(\ell)}] \in \mathbb{R}^{(P+1)\times d}$ from the intermediate representation after the second layer normalization, where $\boldsymbol{z}_0^{(\ell)}$ denotes the `[CLS]` token representation, $\{\boldsymbol{z}_i^{(\ell)}\}_{i=1}^P$ correspond to $P$ patch-level embeddings, and $d$ is the hidden dimension.

To capture both global and spatial information at each layer $\ell$, we extract two complementary representations, which have been shown to improve performance (see Appx. A.10):

$$\boldsymbol{h}_{\text{CLS}}^{(\ell)} = \boldsymbol{z}_0^{(\ell)}; \quad \boldsymbol{h}_{\text{AP}}^{(\ell)} = \frac{1}{P}\sum_{i=1}^{P}\boldsymbol{z}_i^{(\ell)}. \tag{1}$$

The `CLS` token provides a learned global summary while average pooling captures spatial feature statistics. Building on evidence that intermediate layers independently encode valuable task-relevant information (Appx. A.8), we aim to leverage the hierarchical feature evolution across intermediate layers to improve downstream task performance.

Formally, given a subset of layers $\mathcal{L} = \{\ell_1, \ldots, \ell_{|\mathcal{L}|}\} \subseteq \{1, \ldots, L\}$, we stack their representations to form

$$\boldsymbol{H}_{\mathcal{L}} = \begin{bmatrix} \boldsymbol{h}_{\text{CLS}}^{(\ell_1)} & \cdots & \boldsymbol{h}_{\text{CLS}}^{(\ell_{|\mathcal{L}|})} & \boldsymbol{h}_{\text{AP}}^{(\ell_1)} & \cdots & \boldsymbol{h}_{\text{AP}}^{(\ell_{|\mathcal{L}|})} \end{bmatrix}^{\top} \in \mathbb{R}^{2|\mathcal{L}|\times d} \tag{2}$$

The goal is to learn an attention-based fusion function $f_\theta : \mathbb{R}^{2|\mathcal{L}|\times d} \to \mathbb{R}^d$ with learnable parameters $\theta$ that produces an optimal task-specific representation.

## 3.1 ATTENTION-BASED LAYER FUSION

To combine representations, we extend the attentive probing paradigm of Chen et al. (2024) from final-layer patch tokens to the complete set of intermediate layer features.

**Multi-Head Attention Design.** We employ a multi-head cross-attention mechanism that uses the `CLS` and `AP` tokens from intermediate transformer layers as input, rather than all final-layer patches as in prior work. Our method attends over the complete set of intermediate representations $H_{\mathcal{L}}$, enabling task-adaptive selection of optimal abstraction levels.

For each head $m \in \{1, \ldots, M\}$, we introduce trainable projection matrices $\boldsymbol{W}_{\text{key}}^{(m)}, \boldsymbol{W}_{\text{val}}^{(m)}, \boldsymbol{W}_{\text{query}}^{(m)} \in \mathbb{R}^{d\times d_h}$, with head dimensionality $d_h = 2d/M$. The shared learnable query matrix $\boldsymbol{Q} \in \mathbb{R}^{1\times d}$ serves as a task-relevance prototype. It adapts during training to prioritize layers containing task-relevant features. For each head $m$, we compute keys and values from the layer representations $\boldsymbol{H}_{\mathcal{L}}$ and queries from the shared query matrix $\boldsymbol{Q}$:

$$\boldsymbol{K}^{(m)} = \boldsymbol{H}_{\mathcal{L}}\boldsymbol{W}_{\text{key}}^{(m)}, \quad \boldsymbol{V}^{(m)} = \boldsymbol{H}_{\mathcal{L}}\boldsymbol{W}_{\text{val}}^{(m)}, \quad \boldsymbol{Q}^{(m)} = \boldsymbol{Q}\boldsymbol{W}_{\text{query}}^{(m)} \tag{3}$$

The output of each head is computed as:

$$\boldsymbol{h}_{\text{head}}^{(m)} = \text{dropout}\left(\text{softmax}\left(\frac{\boldsymbol{Q}^{(m)}\boldsymbol{K}^{(m)\top}}{\sqrt{d_h}}\right)\right)\boldsymbol{V}^{(m)}, \tag{4}$$

where the attention dropout is used as regularization during training. The fused representation is obtained after a linear transformation of the concatenated heads:

$$\boldsymbol{h}_{\text{fused}} = \left[\boldsymbol{h}_{\text{head}}^{(1)} \oplus \cdots \oplus \boldsymbol{h}_{\text{head}}^{(M)}\right]\boldsymbol{W}_{\text{out}} + \boldsymbol{b}_{\text{out}}. \tag{5}$$

Classification for a downstream task with $K$ classes is performed using a single linear layer with softmax activation $\hat{\boldsymbol{y}} = \text{softmax}(\boldsymbol{W}_{\text{clf}}\boldsymbol{h}_{\text{fused}} + \boldsymbol{b}_{\text{clf}})$. This design keeps the parameter count independent of the number of layers, with the hidden dimension $d$ being the dominant factor and adds minimal computational overhead, requiring only lightweight attention computations over intermediate representations while keeping the pretrained backbone frozen (Appx. A.9). The attention computation scales with $\mathcal{O}\left(|\mathcal{L}|^2\right)$ rather than $\mathcal{O}\left(P^2\right)$ for attentive probes on all patches of the last layer. For ImageNet-sized input images, $P \approx 200$, $L \approx 12$, thus, $|\mathcal{L}| \ll P$ yields an order of magnitude reduction in attention complexity. Rather than manually selecting which layers to include, we find that the use of all intermediate representations $\mathcal{L}_{\text{all}} = \{1, 2, \ldots, L\}$ performs best, as the learned attention weights automatically determine layer relevance.

## 3.2 LINEAR (CONCATENATION-BASED) LAYER FUSION

To validate the effectiveness of adaptive weighting, we additionally consider the naive approach of combining intermediate representations through concatenation, which we refer to as Linear in our plots. This baseline applies a linear classifier directly to the concatenated features:

$$\hat{\boldsymbol{y}} = \text{softmax}\left(\boldsymbol{W}_{\text{clf}}\left[\bigoplus_{\ell \in \mathcal{L}} \boldsymbol{h}_{\text{CLS}}^{(\ell)} \oplus \boldsymbol{h}_{\text{AVG}}^{(\ell)}\right] + \boldsymbol{b}_{\text{clf}}\right). \quad (6)$$

This approach leverages intermediate features but lacks the adaptive weighting capability of our attention-based fusion. The importance of each layer's contribution is learned by the single linear classifier but remains fixed for all inputs after training, whereas the attentive probe, in principle, adapts its weighting.

## 4 EXPERIMENTS

To validate attention-based layer fusion, we conduct a large-scale empirical study asking: (1) Does adaptively fusing intermediate representations outperform strong last-layer baselines? (2) How do learned fusion strategies vary across architectures, training paradigms, and task domains? We observe consistent improvements from using intermediate layers, with the attention mechanism learning effective layer weights for each downstream task.

### 4.1 EXPERIMENTAL SETUP

We first outline our experimental framework, including the probing methods we compare against and our selection of evaluation models and datasets. For reproducibility, we release our code[1] and provide further implementation details in Appx. A.1.

**Probing Methods.** We evaluate probing strategies along three axes: the source layer (intermediate vs. last), the tokens (CLS and/or AP), and the fusion method (linear vs. attentive). To ensure consistency, we denote probes as [layers] ([tokens], [fusion type]). We consider two main baselines: Last layer (CLS, linear), the standard linear probe; and Last layer (all tokens, attentive), an attentive probe (Chen et al., 2024) applying multi-head attention over all $\mathcal{L}_{\text{last}}$ tokens (AAT).

Our primary approach applies attention-based fusion across all layers ($\mathcal{L}_{\text{all}}$). For completeness, we also test concatenation and attention over subsets $\mathcal{L} \in \mathcal{L}_{\text{last}}, \mathcal{L}_{\text{mid+last}}, \mathcal{L}_{\text{quarterly}}, \mathcal{L}_{\text{all}}$. Here, $\mathcal{L}_{\text{mid+last}}$ selects the middle and last ViT layers, and $\mathcal{L}_{\text{quarterly}}$ selects the last layer from each quarter of a ViT.

**Models.** We evaluate nine pretrained ViTs spanning three families (supervised ViTs, self-supervised DINOv2, and image–text aligned CLIP), each available in small, base, and large variants, to study the effects of training objective and model capacity. We freeze the backbone, training only the attention-fusion module and classifier on extracted features. See Appx. A.2 for details. All three families use CLS tokens in their training objectives, which may compress information into this summary representation. We additionally evaluate Masked Autoencoders (He et al., 2022), which avoid such compression, in Appx. A.6.

**Datasets.** Our evaluation covers a diverse suite of 20 datasets from the clip-benchmark (Cherti & Beaumont, 2025) and the Visual Task Adaptation Benchmark (VTAB) (Zhai et al., 2020). We provide details in Appx. Tab. 2.

**Training Objective.** To handle class imbalance, we apply a weighted cross-entropy loss (Aurelio et al., 2019), where the loss for each class is inversely weighted by its sample. This weighting scheme balances learning across minority and majority classes. To reduce overfitting in the probing module, we apply standard regularization techniques including attention dropout, weight decay, and light jittering of intermediate representations (see Appx. A.1).

**Evaluation Metric.** We evaluate model performance using top-1 balanced accuracy on the respective test sets. To enable an intuitive comparison of performances across datasets, we report the absolute accuracy gain (in percentage points [pp]) of each method over the standard linear probe

---

[1]https://anonymous.4open.science/r/intermediate-layer-fusion

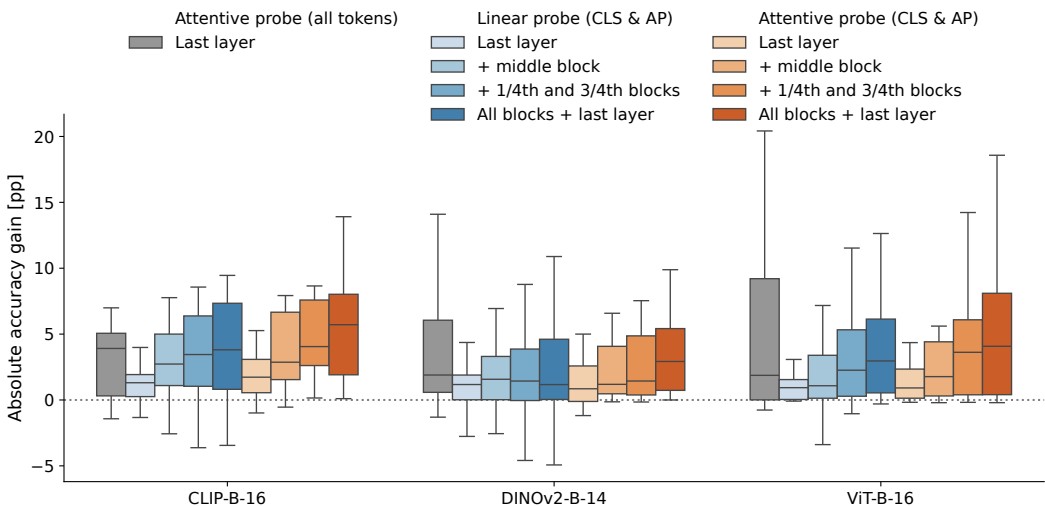

Figure 2: Absolute accuracy gain (percentage points) of linear (blue) and attentive probes (orange) when fusing an increasing number of intermediate layer representations ($\mathcal{L}_{\text{last}}$, $\mathcal{L}_{\text{mid+last}}$, $\mathcal{L}_{\text{quarterly}}$, and $\mathcal{L}_{\text{all}}$), as well as AAT (grey) aggregated across datasets for the three base models. Including more intermediate layers improves for all models, with our attentive probe over all layers achieving the highest median gain and consistently outperforms the simple linear probe (zero line).

`CLS` baseline:

$$\Delta_{\text{Acc}}(\text{method}) = \text{Acc}_{\text{bal}}(\text{method}) - \text{Acc}_{\text{bal}}(\text{CLS}_{\text{linear}}), \tag{7}$$

which is positive if the method outperforms the baseline. A single run is reported for each experiment, as preliminary tests indicate small variance across runs with different random seeds (see Appx. A.15).

## 4.2 THE EFFECT OF INTERMEDIATE LAYERS ON DOWNSTREAM PERFORMANCE

In this section, we aim to assess (1) the impact of adding intermediate layers on the downstream performance compared to using only the final layer, and (2) the performance differences between attentive and linear fusion strategies. To test this, we evaluate the three base models (CLIP-B-16, DINOv2-B-14, and ViT-B-16) and measure the accuracy gain (Eq. 7) relative to the standard linear probe on the `CLS` token as we include more intermediate representations.

Fig. 2 shows that adding representations boosts performance. Both naive concatenation and our attentive fusion significantly benefit from deeper feature pools (p-value $\leq 0.04$, FDR-corrected Wilcoxon)[2], confirming that intermediate layers encode complementary information absent in the final layer's `CLS` and `AP` tokens. However, the two fusion strategies differ substantially in robustness. While concatenation shows positive median gains, it exhibits high variance across tasks, with some datasets experiencing substantial performance degradation. In contrast, our attentive probe on all layers shows the largest (median) performance gains, consistently outperforming the concatenation fusion strategy (p-value $\leq 0.013$, FDR-corrected Wilcoxon)[3], demonstrating its capacity to adaptively emphasize useful layers while ignoring irrelevant ones.

We compare against the attentive probe on all tokens (AAT) in the last layer, which accesses fine-grained semantic as well as spatial information from all patch tokens (incl. `CLS`). AAT proves unstable with high variance and occasional underperformance. Our method, which attends over summary tokens from all layers, achieves higher median gains with markedly less variance. Finally, we find that our observations are robust across different attentive parameterizations of probes, which we analyze in detail in Appx. A.14.

---

[2]Testing "All layers (CLS + AP, attentive)" and "All layers (CLS + AP, linear)" against "Last layer (CLS + AP, attentive)" and "Last layer (CLS + AP, linear)", respectively for all models.

[3]Testing "All layers (CLS + AP, attentive)" against "All layers (CLS + AP, linear)" for all models

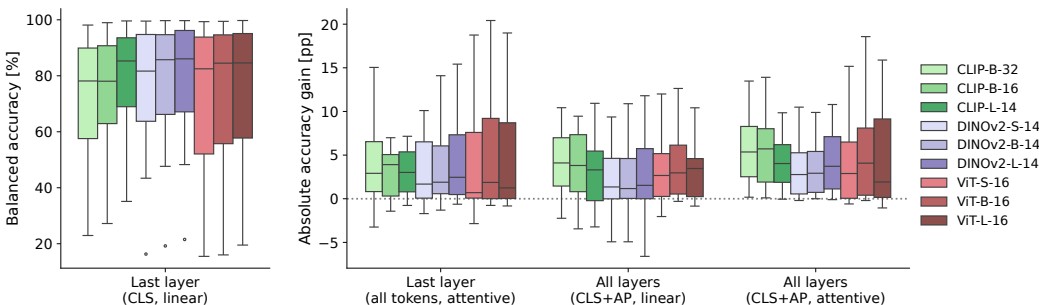

Figure 3: Balanced accuracy distributions of baseline (left panel) and absolute accuracy gains in percentage points (right panel) for different representation fusion methods across model architectures, aggregated over all 20 datasets. The substantial benefits from attentive probing of intermediate layers [All layers (CLS +AP, attentive)] persist even for large models, indicating that large models fail to encode all task-relevant information in their final layer's CLS token.

Together, these findings validate two central claims of our work: (1) intermediate layers contain valuable information for downstream tasks that is not captured by the final layer alone, and (2) an attentive fusion mechanism is crucial to harness this information safely and consistently across diverse downstream tasks.

### 4.3 INTERMEDIATE LAYER FUSION ACROSS MODEL SCALES

To assess whether intermediate-layer fusion depends on model size, we evaluate small, base, and large variants across three model families. Consistent with previous results, adding intermediate layers improves performance at all scales (Fig. 3), with attentive probes outperforming concatenation. Detailed per-dataset results for each model are provided in Appx. Fig. 5. However, the magnitude of these gains varies by training objective, yielding distinct scaling behaviors across families.

CLIP models show the most consistent gains, with smaller models (CLIP-B-32/16) benefiting more than the large model, likely because they fail to distill all relevant information into the final layer.

In contrast, DINOv2 models exhibit the opposite trend: performance gains increase with model size, reflecting richer features throughout the network hierarchy. While the CLS linear probe already substantially improves from DINOv2-S-14 to DINOv2-L-14, attentive fusion adds a further mean gain of 6.04 [pp] for DINOv2-L-14. AAT yields a slightly higher average gain (6.23 [pp]), but suffers from greater instability and lower median gain. This instability likely stems from AAT's reliance on hundreds of patch tokens (257 for DINOv2-S/B/L-14), which amplifies task-specific noise. In contrast, our method aggregates only 24/48 summary tokens, producing more consistent improvements across tasks.

Finally, for supervised ViT models, gains peak at the base architecture. The large variant benefits less proportionally, which could be due to overfitting from the higher hidden dimension or to diminishing returns in representational richness as backbone capacity grows. In either case, while relative improvements shrink, absolute performance still increases when attending across layers.

In summary, attentive fusion consistently improves performance across model sizes. Contrary to the intuition that smaller models would benefit more due to their weaker base performance, we find that larger models obtain equally substantial gains. Highlighting our method's ability to scale with model capacity, it complements rather than replaces the final-layer representation. At the same time, the variability across datasets suggests that the benefits are task dependent, which we elaborate on in the following section.

### 4.4 TASK DEPENDENT BENEFITS OF INTERMEDIATE LAYER FUSION

Tab. 1 summarizes the effect of different probing strategies across the 20 benchmark datasets, averaged over the nine models considered in this work. The strongest gains are achieved by attentively fusing representations from all layers (yielding the highest mean rank).

Table 1: Absolute performance gains (pp) of probing methods relative to baseline CLS token linear probe on the final layer. Results show mean ± standard deviation across all 9 models. **Bold** indicates the largest performance gain per dataset, and underlined indicates the second-largest. Baseline balanced accuracy reported for reference. Dataset categories: Natural multi-domain (MD) images; Natural single domain (SD) images; Specialized (domain-specific imagery); Structured (datasets with structural patterns).

| Category | Dataset | Baseline Bal. accuracy (CLS, linear) | Last layer (all tokens, attentive) | Last layer (CLS + AP, linear) | All layers (CLS+AP, linear) | Last layer (CLS + AP, attentive) | All layers (CLS+AP, attentive) |
|---|---|---|---|---|---|---|---|
| Natural (MD) | STL-10 | 99.29 ± 0.51 | 0.01 ± 0.16 | -0.01 ± 0.12 | 0.03 ± 0.10 | 0.03 ± 0.08 | **0.04 ± 0.17** |
| | CIFAR-10 | 96.91 ± 1.93 | 0.42 ± 0.58 | 0.08 ± 0.11 | 0.61 ± 0.71 | 0.19 ± 0.29 | **0.77 ± 0.79** |
| | Caltech-101 | 95.57 ± 1.40 | 0.23 ± 0.52 | 0.43 ± 0.41 | 0.36 ± 0.63 | 0.09 ± 0.42 | **0.88 ± 0.77** |
| | PASCAL VOC 2007 | 87.82 ± 2.31 | -0.22 ± 1.24 | 1.38 ± 0.49 | **1.46 ± 0.99** | 1.19 ± 0.88 | 1.24 ± 0.89 |
| | ImageNet-1k | 81.40 ± 4.49 | 0.85 ± 1.43 | 0.33 ± 0.46 | 0.99 ± 1.75 | 0.15 ± 0.62 | **1.24 ± 1.62** |
| | CIFAR-100 | 85.45 ± 5.71 | 1.73 ± 1.33 | 0.61 ± 0.21 | 2.76 ± 2.48 | 0.87 ± 0.56 | **3.33 ± 2.75** |
| | Country-211 | 21.48 ± 6.35 | -0.83 ± 1.66 | 1.18 ± 0.54 | 3.26 ± 1.05 | 1.35 ± 0.65 | **4.96 ± 1.37** |
| Natural (SD) | Pets | 93.98 ± 2.36 | -0.23 ± 0.83 | -0.05 ± 0.41 | -2.01 ± 1.04 | 0.12 ± 0.53 | **0.29 ± 0.76** |
| | Flowers | 98.03 ± 2.60 | 0.41 ± 0.93 | 0.40 ± 0.75 | -0.25 ± 0.57 | 0.06 ± 0.76 | **0.46 ± 0.97** |
| | Stanford Cars | 77.81 ± 10.65 | **8.97 ± 5.22** | 0.50 ± 1.07 | -0.86 ± 3.76 | 1.97 ± 1.95 | 6.35 ± 3.71 |
| | FGVC Aircraft | 55.69 ± 12.18 | **9.27 ± 4.37** | -0.96 ± 2.22 | -1.62 ± 5.01 | 1.84 ± 2.09 | 6.43 ± 3.25 |
| | GTSRB | 71.51 ± 7.46 | **18.02 ± 6.37** | 4.23 ± 2.60 | 8.76 ± 4.20 | 4.69 ± 2.41 | 13.47 ± 4.92 |
| | SVHN | 56.06 ± 5.91 | **30.31 ± 5.08** | 6.94 ± 2.59 | 24.40 ± 4.41 | 7.39 ± 3.70 | 27.25 ± 4.24 |
| Specialized | PCAM | 82.04 ± 2.15 | 5.03 ± 1.47 | 1.38 ± 0.56 | **5.32 ± 1.62** | 2.66 ± 1.33 | 2.85 ± 2.53 |
| | EuroSAT | 93.89 ± 2.52 | 3.38 ± 2.18 | 1.65 ± 1.17 | 4.08 ± 2.48 | 1.82 ± 1.22 | **4.37 ± 2.41** |
| | RESISC45 | 90.45 ± 1.69 | 4.07 ± 1.05 | 1.32 ± 0.74 | 4.53 ± 0.99 | 1.82 ± 0.59 | **5.23 ± 1.10** |
| | Diabetic Retinopathy | 45.80 ± 2.46 | 1.94 ± 1.90 | 1.55 ± 0.44 | 5.92 ± 2.03 | 1.86 ± 0.77 | **6.86 ± 2.00** |
| Structured | DTD | 75.99 ± 3.47 | 1.41 ± 2.19 | 1.18 ± 1.76 | 4.04 ± 2.19 | 2.53 ± 1.67 | **4.05 ± 1.92** |
| | FER2013 | 59.08 ± 4.61 | 7.74 ± 2.15 | 2.18 ± 1.05 | 6.25 ± 1.19 | 3.61 ± 1.13 | **10.05 ± 1.76** |
| | Dmlab | 44.91 ± 3.49 | **13.69 ± 2.77** | 1.81 ± 0.45 | 7.92 ± 1.95 | 2.61 ± 1.65 | 10.68 ± 2.78 |
| | Mean rank | - | 2.75 | 4.30 | 2.80 | 3.70 | **1.45** |

The exact magnitude of the improvements from intermediate layers is somewhat task-dependent. On natural multi-domain datasets (CIFAR-10, STL-10), the baseline accuracy is near saturation, and fusion therefore yields relatively small but still significant gains. Fine-grained natural-image tasks (Stanford Cars, FGVC Aircraft, GTSRB, SVHN) benefit most from attentive probing, with gains of 6–30 [pp]. These datasets require subtle distinctions between visually similar categories or precise spatial reasoning, which the final CLS token, optimized for global summarization, tends to suppress. While AAT surpasses our method on these particular tasks by leveraging fine-grained spatial cues from patch embeddings, our approach remains the second-best in almost all cases and, importantly, provides the best average performance and best average rank across all 20 datasets (Tab. 1). Our attentive fusion, relying on aggregated patch information, may miss some subtle spatial details but remains far more stable than AAT and substantially outperforms standard linear probing, highlighting the value and robustness of distributed intermediate features.

Domain-specialized (satellite or medical imagery) and structured datasets (textures, facial expressions, synthetic environments) benefit substantially from including intermediate layers, reflecting the transferability of mid-level features to novel domains and their encoding of compositional patterns. A notable exception is DMLab, where patch-level aggregation performs better, because fine spatial detail is critical for this task.

Beyond mean performance gains, stability matters. Attending to all last-layer tokens can excel on certain fine-grained tasks but is brittle, sometimes degrading performance when the CLS token already suffices (e.g., Pets). In contrast, our attention over summary tokens from all layers consistently delivers performance gains across all datasets. Only for PCAM and PASCAL VOC 2007, the linear combination of intermediate layers outperforms the attentive weighting, likely due to overfitting as discussed in Appx. A.11.

Taken together, these results demonstrate that the usefulness of intermediate features varies by task. The benefits are greatest for datasets outside the pretraining domain, where the CLS token alone often proves to be insufficient. While outliers such as PCAM reveal the risk of overfitting, adaptive fusion remains the most reliable strategy for exploiting task-specific signals from intermediate layers.

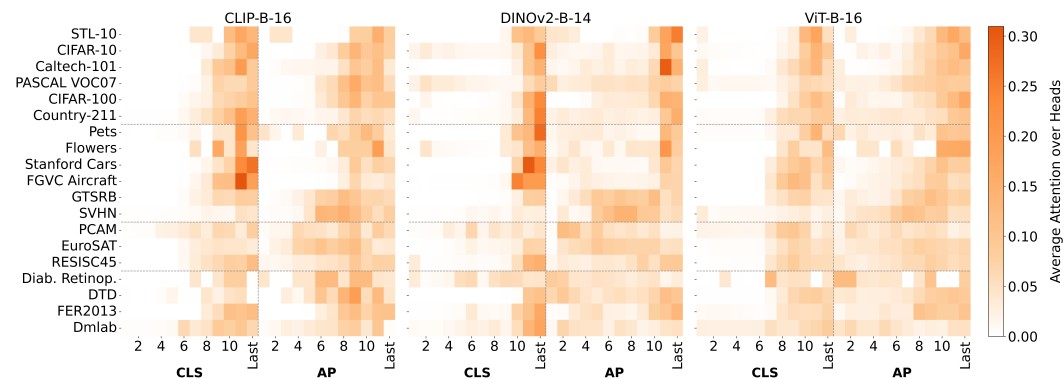

Figure 4: Attention weights across layers and datasets for base models, averaged over heads and samples, are distributed across multiple layers, demonstrating their relevance for downstream tasks.

### 4.5 ANALYZING ADAPTIVE LAYER SELECTION

To understand how our approach adapts to different downstream tasks, we analyze the attention weights of intermediate layers. These weights reveal which layer's representations are most crucial for a given dataset. By aggregating over the attention heads and data samples, the heatmaps indicate how much each layer contributes to the fused representation (Fig. 4 for base and Appx. Fig. 6 for small/large model sizes).

Early layers' `CLS` tokens receive little attention, which is expected since the global summary only becomes semantically rich in later layers. In contrast, average-pooled representations are used across a much wider range of layers. This confirms our hypothesis that spatial averaging preserves valuable textural and structural information throughout the network, complementing the highly processed `CLS` tokens.

Attention distribution varies by dataset. For tasks similar to pretraining, like CIFAR or Pets, attention is high on the last layers' `CLS` and `AP` tokens, as these abstract features are directly useful. In contrast, for tasks that differ from pretraining, such as EuroSAT and FER2013, attention shifts to intermediate layers and their `AP` tokens, consistent with the largest performance gains observed on these datasets. As shown in Appx. A.7 and A.8, intermediate layers alone can achieve comparable performance to the last layer, despite having dissimilar representations. This suggests that these layers provide potential complementary, non-redundant information across layers. Overall, the heatmap confirms that adaptive fusion effectively leverages these lower-level features that might otherwise be lost in the last layers.

## 5 DISCUSSION

The field has long hold the belief that most, if not all, task-relevant information is encoded in the last layers of a neural network model (cf. Devlin et al., 2019; Zhai et al., 2020; Dosovitskiy et al., 2021; Radford et al., 2021; Kornblith et al., 2021; Raghu et al., 2021) and, hence, gravitated toward using the penultimate or final layer for adapting model representations via linear probing (Alain & Bengio, 2017; Kornblith et al., 2019b; Muttenthaler et al., 2023). However, there has recently been suggestive evidence that information relevant for successfully deploying a model downstream may be distributed across several tokens and layers (Oquab et al., 2024; Tu et al., 2023; Chen et al., 2024).

Here, we provide further evidence that intermediate layers in ViTs encode relevant task-specific signals that the `CLS` representation of the final layer does not capture alone. In a supplementary analysis, we find that intermediate layers perform comparably to last layers on certain datasets despite having dissimilar representations, suggesting they hold complementary knowledge. Our attention mechanism allocates significant weights to both intermediate and last layers, indicating intermediate representations contribute meaningful information for downstream predictions. The learned attention weights show that specialized domains like medical and satellite imaging rely heavily on information encoded in intermediate layers, whereas natural image tasks focus on last-layer se-

mantics. We demonstrate that probing via cross-attention, rather than simple affine transformations, effectively leverages intermediate layer representations, and show these benefits hold robustly across different attentive probing architectures.

While standard linear probing becomes unstable when naively extended to multiple layers, our attentive probing mechanism consistently provides improvements across 20 datasets. Although attention over all tokens from the last layer can be highly performative on tasks where precise spatial information is required, it proves brittle with high variance across datasets, making intermediate layers with compact summary tokens a more robust choice for reliable improvements, especially if knowledge about the downstream task is limited. This distinction reflects orthogonal design choices: hierarchical aggregation across layers versus spatial aggregation across patches. For models with `CLS`-focused pretraining (CLIP, DINOv2, supervised ViTs), our hierarchical fusion using summary tokens (`CLS +AP`) is sufficient: average pooling provides spatial statistics to complement `CLS` semantics (Fig. 12), while maintaining stability across tasks. Supplementary experiments with Masked Autoencoders (Appx. A.6), whose pretraining is not `CLS`-based, show that patch-centric models also benefit from hierarchical aggregation, though direct spatial attention (AAT) becomes more advantageous when information remains distributed across individual patches. First experiments (Fig.11) on combining these orthogonal fusion approaches show superior performance, suggesting that incorporating information across the different model's layers is a viable and robust approach for improving downstream adaptation.

**Limitations.** Our attentive probe's token selection strategy (`CLS +AP`) is optimized for models with `CLS`-focused pretraining; spatial averaging may neglect localization cues critical for tasks requiring precise spatial reasoning. Additionally, its greater expressivity introduces additional computational and memory overhead compared to using only the final output token, and can increase overfitting risk, requiring careful regularization. In addition, the spatial averaging used to summarize the remaining tokens may neglect fine-grained spatial details that some tasks require, in particular those necessitating precise localization, where patch-level representations may be more suitable.

**Outlook.** The findings of this paper are in accordance with similar discoveries in language models, where intermediate layers can outperform final representations (Liu et al., 2019; Skean et al., 2025). Together, results across vision and language domains suggest that adaptive access to intermediate representations represents a fundamental principle for the successful deployment of foundation models. This principle extends naturally to emerging biological foundation models for sequences (Brixi et al., 2025), genomics (Theodoris et al., 2023; Schaar et al., 2024), and proteins (Lin et al., 2023), where specialized tasks may benefit from intermediate representations that final layers abstract away. As foundation models proliferate across domains, principled methods to access their full representational hierarchy could prove increasingly valuable for maximizing their utility.

ETHICS STATEMENT

This work adheres to the ICLR Code of Ethics. Our study focuses on probing and adaptation methods for vision transformers using publicly available benchmark datasets from the VTAB and clip-benchmark. No human subjects, private data, or personally identifiable information were used. The datasets we rely on are widely adopted in the vision community, and our experiments follow their respective licenses and usage guidelines. The proposed methods do not pose foreseeable risks of misuse beyond standard applications of image classification. We are committed to transparency and reproducibility, and release code to facilitate verification and further research.

REPRODUCIBILITY STATEMENT

We provide extensive details to ensure reproducibility of our results. The main paper gives an overview of the experimental setup in Sec. 4.1, with further implementation details, including feature extraction, training protocols, hyperparameter search, and regularization strategies, provided in Appx. A.1. Dataset descriptions are given in Appx. Tab. 2. We release our full code at `https://anonymous.4open.science/r/intermediate-layer-fusion` to enable exact replication of our experiments.

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

# A  APPENDIX

## A.1  IMPLEMENTATION DETAILS

This section describes the technical implementation approach and experimental setup used to evaluate the attention-based intermediate layer fusion mechanisms.

A frozen backbone strategy was adopted, training only the attention fusion mechanism and classification head on top of pre-extracted features. The latent representations (of intermediate and last layers) for each model-dataset combination were extracted using the Python package thingsvision (Muttenthaler & Hebart, 2021), and the experiment code was built on top of the code from Ciernik et al. (2025). Input images were resized to 256px and center-cropped to 224px before applying the model-specific normalizations from the pre-training. Extracted features were then L2-normalized to yield comparable magnitudes. To handle models with varying feature dimensions across layers (e.g., CLIP), we ensured dimensional consistency through zero-padding.

All models were trained for at least 40 epochs using AdamW optimization with cosine annealing learning rate scheduling and a batch size of at most 2048. For small datasets, we adjusted the batch sizes to ensure at least 5 batches per epoch, and increased the number of epochs to guarantee at least 1000 gradient update steps.

To address class imbalance, we trained with a weighted cross-entropy objective (Aurelio et al., 2019), scaling each class by the inverse of its frequency. The loss is $\text{Loss}(y, \hat{y}) = -\frac{1}{N} \sum_{j=1}^{N} \sum_{i=1}^{K} w_i y_{ji} \log \hat{y}_{ji}$, where $y_{ji}$ is the one-hot ground-truth label for sample $j$ and class $i$, and $\hat{y}_{ji}$ is the predicted probability. $w_i$ are class weights computed as $w_i = \frac{N}{K \cdot n_i}$, with $N$ being the total number of training samples, $K$ the number of classes, and $n_i$ the number of samples in class $i$. This weighting balances learning across minority and majority classes.

Hyperparameter selection used a stratified 80/20 train-validation split with grid search over learning rates $\{0.1, 0.01, 0.001\}$, attention dropout rates $\{0.0, 0.1, 0.3\}$, and weight decay values $\{10^{-6}, 10^{-5}, 10^{-4}, 0.001, 0.01, 0.1, 1.0\}$, except for the AAT baseline, where we used the reported weight decay of 0.1 Chen et al. (2024). We selected the combination that achieved the best validation balanced accuracy.

To prevent overfitting, we applied gradient norm clipping at 5.0 and added Gaussian noise $\mathcal{N}(0, 0.05)$ to representations with probability 0.5 during training.

For the representation-fusion attention mechanism, we adjust the number of heads to match the number of representations being fused (cf. Appx. A.12). For example, when fusing CLS and AP tokens from all 12 layers of a ViT-B-16 model, we used $M = 24$ heads. For the AAT baseline, we used 8 attention heads following Chen et al. (2024), as increasing the number of heads did not yield substantial improvements. The learned query tokens were initialized from a normal distribution $\mathcal{N}(0, 0.02)$.

## A.2 Model Details

This section provides the specific model variants and patch sizes used in our experiments across three model families: supervised ViTs, self-supervised DINOv2 models, and image-text aligned CLIP models.

- **Supervised ViT:** ViT-S/16, ViT-B/16, and ViT-L/16 pretrained on ImageNet-21K and fine-tuned on ImageNet-1K  (Deng et al., 2009; Ridnik et al., 2021).
- **Self-Supervised DINOv2:** ViT-S-14, ViT-B-14, and ViT-L-14 , pretrained on the LVD-142M dataset (Oquab et al., 2024).
- **Image-Text Alignment CLIP:** OpenCLIP models ViT-B-32, ViT-B-16, and ViT-L-14 (Cherti et al., 2023; Ilharco et al., 2021)) following the CLIP architecture and using its pretrained weights (Radford et al., 2021)). As a small-capacity CLIP model, we use ViT-B/32; its larger patch size significantly reduces the number of input tokens, making its computational and representational capacity analogous to the "Small" variants in the other families.

## A.3 Dataset Details

Table 2: Overview of the 19 datasets used in our experiments including the size of both train and test set, number of classes, and the Class Imbalance Ratio (CIR) calculated by $\frac{N_{\text{Majority Class}}}{N_{\text{Minority Class}}}$.

| Category | Dataset | Train Size | Test Size | Classes | CIR | Reference |
|---|---|---|---|---|---|---|
| Natural (MD) | STL-10 | 5 000 | 8 000 | 10 | 1 | Coates et al. (2011) |
| | CIFAR-10 | 45 000 | 10 000 | 10 | 1.02 | Krizhevsky (2009) |
| | Caltech-101 | 2 753 | 6 085 | 102 | 1.3 | Fei-Fei et al. (2006) |
| | PASCAL VOC 2007 | 7 844 | 14 976 | 20 | 20.65 | Everingham et al. (2010) |
| | CIFAR-100 | 45 000 | 10 000 | 100 | 1.06 | Krizhevsky (2009) |
| | Country-211 | 31 650 | 21 100 | 211 | 1 | Radford et al. (2021) |
| Natural (SD) | Pets | 2 944 | 3 669 | 37 | 1.24 | Parkhi et al. (2012) |
| | Flowers | 1 020 | 6 149 | 102 | 1 | Nilsback & Zisserman (2008) |
| | Stanford Cars | 8 144 | 8 041 | 196 | 2.83 | Krause et al. (2013) |
| | FGVC Aircraft | 3 334 | 3 333 | 100 | 1.03 | Maji et al. (2013) |
| | GTSRB | 26 640 | 12 630 | 43 | 10 | Stallkamp et al. (2012) |
| | SVHN | 65 931 | 26 032 | 10 | 2.98 | Netzer et al. (2011) |
| Specialized | PCAM | 262 144 | 32 768 | 2 | 1 | Veeling et al. (2018) |
| | EuroSAT | 16 200 | 5 400 | 10 | 1.58 | Helber et al. (2019) |
| | RESISC45 | 18 900 | 6 300 | 45 | 1.16 | Cheng et al. (2017) |
| | Diabetic Retinopathy | 35 126 | 42 670 | 5 | 36.45 | Dugas et al. (2015) |
| Structured | DTD | 1 880 | 1 880 | 47 | 1 | Cimpoi et al. (2014) |
| | FER2013 | 28 709 | 7 178 | 7 | 16.55 | Goodfellow et al. (2015) |
| | Dmlab | 65 550 | 22 735 | 6 | 1.98 | Zhai et al. (2020) |

An overview of all datasets used in this work is given in Tab. 2. Following VTAB (Zhai et al., 2020), the datasets are categorized by domain. We separate natural images into multi-domain (MD) and single-domain (SD) datasets, and include specialized as well as structured datasets.

## A.4 Downstream performance for all datasets and models

We present the balanced test accuracies across all 19 downstream datasets for each of our nine models. Each table (Figures 5a, 5b, and 5c) shows four different probing configurations: (1) last layer `CLS` token with linear probing, (2) last layer all tokens with attentive probing, (3) all layers `CLS` and `AP` token with linear probing, and (4) all layers `CLS` and `AP` token with attentive probing.

The bottom rows of each table report summary statistics of the absolute performance gains relative to the baseline last-layer CLS linear probe, including the minimum, median, maximum, mean, and standard deviation of improvements across all datasets. Color coding indicates relative performance within each model family, with darker colors representing better performance.

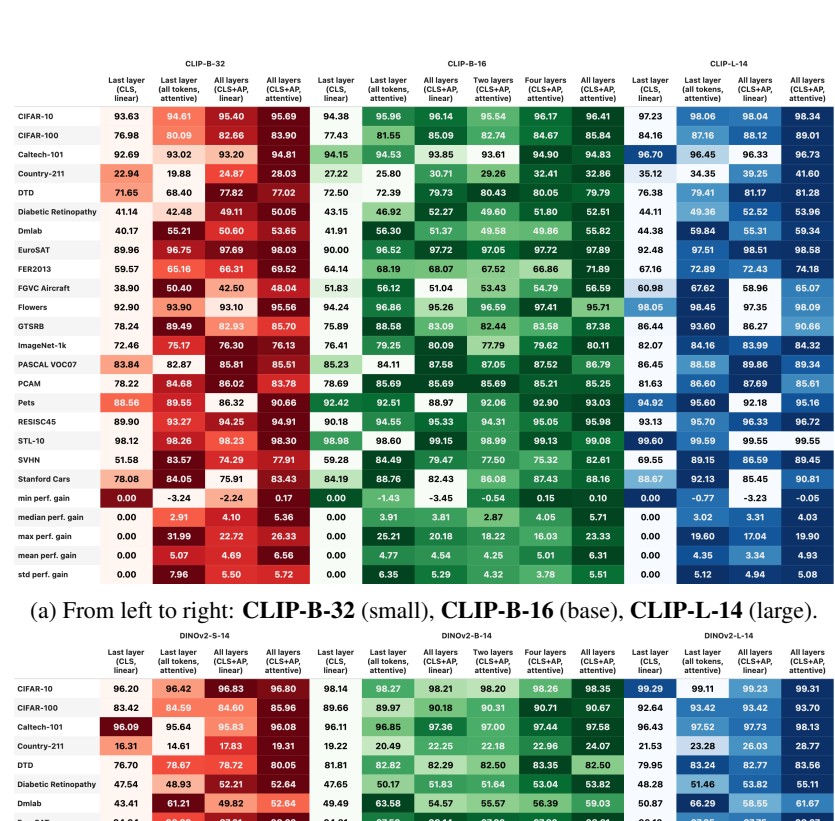

(a) From left to right: **CLIP-B-32** (small), **CLIP-B-16** (base), **CLIP-L-14** (large).

(b) From left to right: **DINOv2-S-14** (small), **DINOv2-B-14** (base), **DINOv2-L-14** (large).

(c) From left to right: **ViT-S-16** (small), **ViT-S-16** (base), **ViT-L-16** (large).

Figure 5: Accuracies per model and dataset

## A.5 ATTENTION HEATMAPS FOR SMALL AND LARGE MODELS

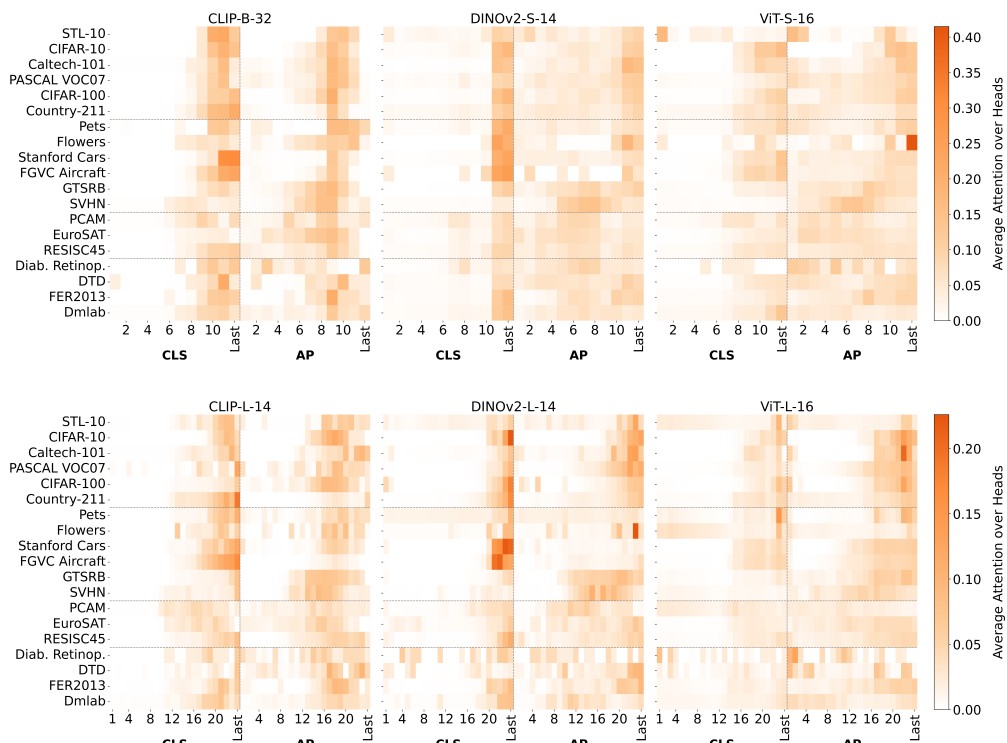

Figure 6: Aggregated Attention maps from our attentive probe for small (top) and large (bottom) models. Attention patterns vary more with the dataset than with model scale, underscoring the task-dependent relevance of intermediate layer features.

Fig. 6 compares the aggregated attention across our small and large models. Despite substantial differences in scale and twice as many layers for the large models, the attention patterns are very similar. This underlies our intuition that the relevance of intermediate layers depends more on the task characteristics than on model size or objective, which seem to learn very similar hierarchies. Specialized and structural datasets drive attention toward intermediate layers, while natural image datasets close to the pre-training domain rely more on the later-layer `CLS` tokens. Notably, in cases like GTSRB and SVHN, where linear `CLS` probing fails but our method achieves large gains, the probe shifts attention to the `AP` tokens. These results reinforce that our mechanism adapts flexibly to task demands while remaining consistent across models of very different scales and pre-training objectives.

## A.6 ADDITIONAL EXPERIMENTS WITH MASKED AUTO ENCODER

Masked Autoencoders (MAEs) (He et al., 2022) represent a distinct class of pretrained models whose representational structure differs fundamentally from that of CLIP, DINOv2, or supervised ViTs, as they are trained exclusively via patch-level reconstruction and thus do not use summary tokens in their loss. Prior work (Przewieźlikowski et al., 2025) has shown that MAEs retain highly localized information until the final layers and therefore benefit most from probes that attend over all patch tokens rather than relying on summary tokens. We confirm this observation: for both MAE-B-16 and MAE-L-16, an attentive probe operating over all tokens achieves the strongest overall performance (Fig. 7 and Fig. 8). Nevertheless, our intermediate-layer attentive fusion, which operates only on the aggregated `CLS`/`AP` tokens, still produces large gains over last-layer `AP` probes, with mean improvements of 22.4 (base) or 24.7 (large) percentage points. In most datasets, it ranks second only to the full-token attentive probe, and on two datasets, it even surpasses it. This demon-

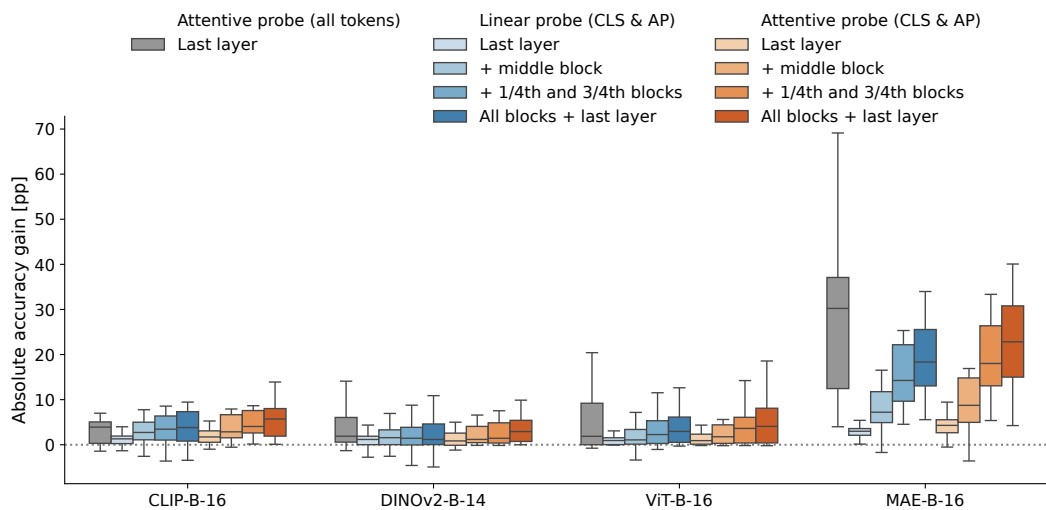

Figure 7: Absolute accuracy gain (percentage points) of linear (blue) and attentive probes (orange) when fusing an increasing number of intermediate layer representations ($\mathcal{L}_{\text{last}}$, $\mathcal{L}_{\text{mid+last}}$, $\mathcal{L}_{\text{quarterly}}$, and $\mathcal{L}_{\text{all}}$), as well as AAT (grey) aggregated across datasets for the three base models as well as the **base MAE model**. For MAE, the simple linear probe on `AP` tokens is insufficient for most tasks, explaining the large gain in accuracy by either including spatial (all tokens last layer) or hierarchical (`CLS` & `AP`, all blocks) information.

strates that layer-wise fusion recovers complementary information across depth, highlighting the orthogonality between token aggregation (spatial dimension) and layer aggregation (hierarchical dimension). These results illustrate an important representational difference. MAEs do not compress information into the `CLS` token, so probes that access all patch tokens are inherently favored. Nevertheless, when restricted to summary tokens, multi-layer fusion substantially mitigates this limitation. Thus, our results on MAE confirm that performance depends both on how information is distributed across tokens and how it evolves across layers. Our proposed layer-fusion approach remains effective, especially when a model exposes meaningful layerwise summary representations.

To better understand these results, we inspect the learned attention patterns for MAE-B-16 and MAE-L-16 (Fig. 9). Since the MAE `CLS` token is not trained, the probe naturally places nearly all its attention on the `AP` tokens, confirming that summary representations are weak in this model family. The attention also concentrates on the later layers, consistent with the fact that MAEs preserve spatial detail until the end of the network and perform little semantic compression. As a result, aggregating information from intermediate `AP` tokens enables our fusion to recover much of the depth-wise structure, allowing it to approach the performance of probes with access to all spatial tokens.

## A.7 RELATIONSHIP BETWEEN INTERMEDIATE-LAYER PERFORMANCE AND REPRESENTATIONAL SIMILARITY

Prior work has shown that intermediate layers contain task-relevant information accessible via linear probing (Alain & Bengio, 2017). Following Kornblith et al. (2019a), we examine the relationship between downstream performance and representational similarity measured by Centered Kernel Alignment (CKA) with RBF kernel ($\sigma = 0.2$), which emphasizes local neighborhood similarities relative to the final layer's representation. To study this across architectures and feature types, we trained linear probes on all intermediate layers of the four base models (CLIP-B-16, DINOv2-B-14, ViT-B-16, MAE-B-16) on CIFAR-100, GTSRB, FER2013, and EuroSAT. For all models except MAE, we probe the `CLS` token, while for MAE, we use the `AP` token.

Fig. 10 shows that CKA similarity to the final layer is not strongly predictive of downstream performance. While similarity tends to increase rapidly in the later layers, the largest accuracy gains often occur in early or middle layers. Notably, even though these intermediate representations are

| | MAE-B-16 | | | | | | | MAE-L-16 | | | | |
| | Last layer (CLS, linear) | Last layer (AP, linear) | Last layer (all tokens, attentive) | All layers (CLS+AP, linear) | Two layers (CLS+AP, attentive) | Four layers (CLS+AP, attentive) | All layers (CLS+AP, attentive) | Last layer (CLS, linear) | Last layer (AP, linear) | Last layer (all tokens, attentive) | All layers (CLS+AP, linear) | All layers (CLS+AP, attentive) |
|---|---|---|---|---|---|---|---|---|---|---|---|---|
| CIFAR-10 | 50.07 | 57.73 | 92.85 | 84.47 | 73.22 | 83.84 | 88.32 | 69.70 | 66.21 | 95.62 | 93.49 | 94.33 |
| CIFAR-100 | 26.03 | 32.90 | 75.91 | 66.36 | 49.13 | 66.25 | 71.24 | 40.93 | 40.65 | 82.53 | 79.01 | 79.95 |
| Caltech-101 | 62.65 | 71.17 | 93.97 | 87.68 | 67.57 | 88.15 | 91.34 | 75.90 | 73.78 | 95.12 | 91.05 | 94.10 |
| Country-211 | 4.51 | 6.06 | 10.05 | 11.63 | 10.45 | 12.94 | 13.45 | 5.78 | 6.68 | 11.00 | 15.09 | 16.18 |
| DTD | 45.16 | 57.55 | 68.24 | 68.94 | 63.99 | 71.33 | 70.64 | 55.59 | 62.39 | 70.85 | 74.63 | 73.35 |
| Diabetic Retinopathy | 36.39 | 40.74 | 47.45 | 49.55 | 47.00 | 47.09 | 50.13 | 34.98 | 41.26 | 44.82 | 50.95 | 51.29 |
| Dmlab | 27.12 | 29.65 | 62.04 | 43.25 | 35.26 | 40.58 | 46.47 | 29.30 | 30.58 | 65.43 | 46.36 | 49.87 |
| EuroSAT | 75.06 | 81.50 | 98.00 | 96.23 | 93.76 | 96.34 | 97.50 | 79.42 | 82.36 | 98.12 | 97.42 | 97.93 |
| FER2013 | 28.18 | 32.77 | 62.23 | 51.60 | 38.92 | 50.75 | 56.51 | 33.79 | 39.79 | 66.73 | 58.86 | 63.35 |
| FGVC Aircraft | 9.78 | 9.30 | 64.78 | 30.32 | 20.34 | 27.36 | 34.61 | 11.82 | 11.37 | 73.01 | 40.71 | 46.31 |
| Flowers | 44.10 | 59.38 | 93.15 | 83.36 | 76.29 | 87.21 | 88.08 | 59.21 | 63.75 | 94.11 | 88.35 | 91.24 |
| GTSRB | 20.15 | 32.38 | 97.61 | 66.36 | 48.41 | 62.19 | 72.45 | 23.83 | 35.97 | 98.67 | 74.94 | 80.86 |
| ImageNet-1k | 27.78 | 35.64 | 69.05 | 65.03 | 51.98 | 62.83 | 68.10 | 41.06 | 38.64 | 74.20 | 75.25 | 76.86 |
| PASCAL VOC07 | 50.02 | 61.38 | 74.42 | 75.97 | 63.36 | 76.33 | 77.01 | 58.19 | 63.87 | 80.17 | 82.27 | 81.92 |
| PCAM | 73.14 | 77.21 | 85.81 | 83.66 | 81.23 | 82.58 | 81.46 | 80.42 | 78.45 | 87.27 | 85.08 | 81.42 |
| Pets | 41.08 | 59.83 | 90.83 | 77.68 | 63.83 | 78.57 | 83.28 | 63.31 | 57.06 | 92.69 | 85.62 | 89.37 |
| RESISC45 | 53.42 | 70.57 | 94.17 | 90.92 | 85.17 | 90.65 | 92.77 | 62.26 | 72.99 | 94.27 | 93.16 | 93.82 |
| STL-10 | 80.87 | 86.54 | 96.86 | 95.40 | 91.69 | 94.96 | 95.89 | 89.40 | 82.31 | 98.46 | 98.11 | 98.25 |
| SVHN | 32.47 | 34.49 | 93.20 | 65.92 | 46.42 | 65.87 | 73.82 | 37.87 | 35.94 | 94.39 | 76.48 | 82.43 |
| Stanford Cars | 8.08 | 11.81 | 80.93 | 36.96 | 24.01 | 36.53 | 43.21 | 12.30 | 14.28 | 86.34 | 52.01 | 63.02 |
| min perf. gain | -18.75 | 0.00 | 3.99 | 5.57 | -3.59 | 5.37 | 4.25 | -12.14 | 0.00 | 3.56 | 6.63 | 2.98 |
| median perf. gain | -6.66 | 0.00 | 30.23 | 18.34 | 8.74 | 18.02 | 22.82 | -1.09 | 0.00 | 28.18 | 19.62 | 22.19 |
| max perf. gain | 0.48 | 0.00 | 69.13 | 33.98 | 16.91 | 33.35 | 40.07 | 7.09 | 0.00 | 72.06 | 40.54 | 48.74 |
| mean perf. gain | -7.63 | 0.00 | 30.15 | 19.14 | 9.17 | 18.69 | 22.39 | -1.66 | 0.00 | 30.27 | 23.03 | 25.38 |
| std perf. gain | 5.36 | 0.00 | 19.83 | 8.89 | 5.86 | 8.71 | 10.90 | 5.21 | 0.00 | 20.38 | 11.03 | 13.39 |

Figure 8: Detailed results of both base and large MAE on all datasets. While attending over intermediate layer provides already a large benefit, the aggregation over all tokens is a necessity for masked image modeling confirming that the information is distributed over image tokens.

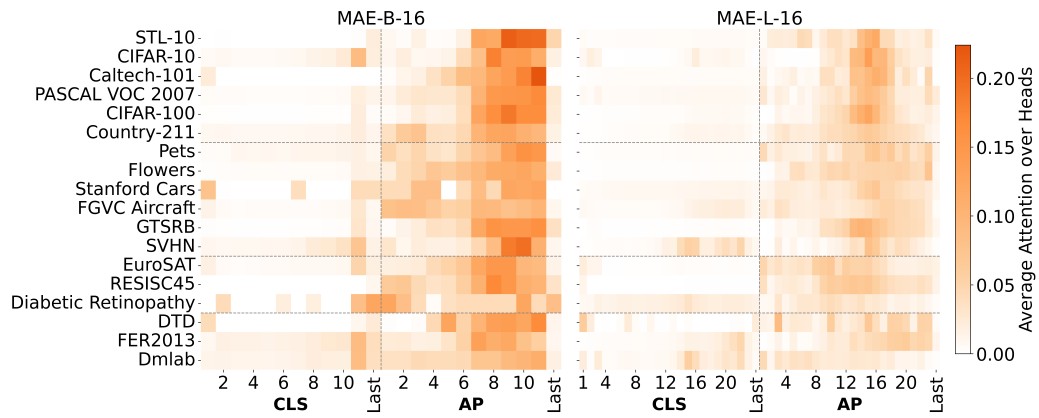

Figure 9: Aggregated intermediate-layer attention maps for MAE-B-16 and MAE-L-16 show that MAEs store task-relevant information predominantly in later `AP` tokens.

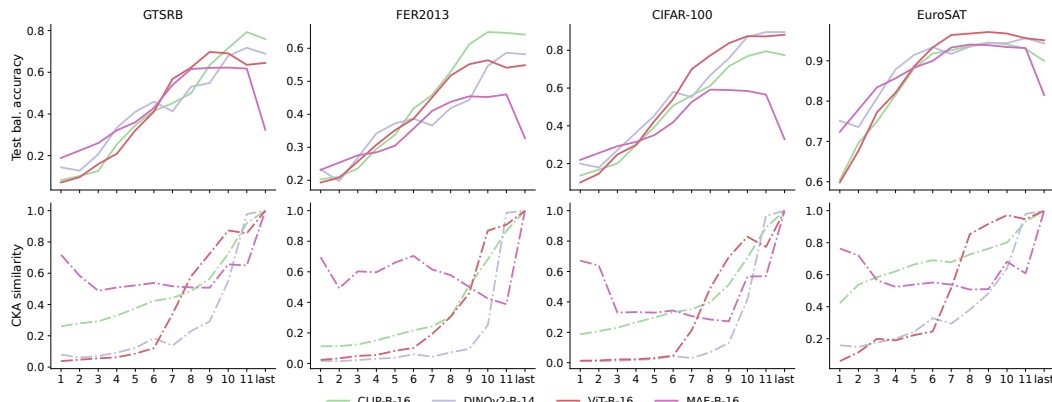

Figure 10: Downstream performance vs. representational similarity across intermediate layers. **Top row**: test balanced accuracy of linear probes on layers 1-11 and the final layer. **Bottom row**: CKA similarity between each layer and the final layer. Intermediate layers can achieve high performance despite low similarity to the final layer.

dissimilar to the final layer, they achieve similar or higher performance, for datasets like GTSRB and EuroSAT, the performance even peaks at layer 6-8.

These results suggest that intermediate layers capture complementary features that are not redundant with the final-layer representations, motivating adaptive fusion strategies to leverage this diverse information effectively.

### A.8 COMPARING PER-LAYER LINEAR AND ATTENTIVE PROBE PERFORMANCE

In this section, we contrast three per-layer probing strategies, linear probing on the `CLS` and `AP` tokens, and an attentive probe that aggregates all tokens of a layer, with our multi-layer attentive fusion, which operates only on the aggregated `CLS` and `AP` representations. Additionally, we add a specialized hybrid probe as discussed below. Results for four datasets and the base models are shown in Fig. 11.

Across all settings, the per-layer attentive probe substantially outperforms linear probes on `CLS` or `AP` tokens, indicating that intermediate-layer information is distributed across spatial tokens and cannot entirely be recovered from a single summary embedding. Despite operating under the stricter constraint of using only `CLS` and `AP` tokens, our multi-layer fusion often exceeds the best per-layer attentive probe. By combining complementary information across depth, our multi-layer attentive fusion recovers much of the signal lost in token aggregation.

Two cases deviate from this trend. For GTSRB, performance peaks in shallow layers and is driven by highly localized features that are not preserved in `CLS` or `AP` tokens, making full-token attention inherently stronger. For MAE, patch tokens encode rich, localized structure from reconstruction training, whereas the `CLS` token receives no explicit supervision to serve as a global summary. Thus, average pooling discards information that an attentive probe over all tokens can utilize. These behaviors are expected given the architectural differences and highlight that, even under strong token-aggregation constraints, our fusion method consistently outperforms all its per-layer components by combining complementary information across layers. To further validate this orthogonality between hierarchical and spatial aggregation, we introduce a hybrid attention probe that combines all tokens from layers $3, 6, 9$ and the last layer. To stabilize training with this larger token set (788 for CLIP, ViT, MAE, and 1028 for DINOv2), we increase attention dropout to $0.5$ and the number of heads to $24$, leaving all other hyperparameters untouched. This hybrid probe consistently outperforms both AAT applied to the last layer and our intermediate layer fusion relying only on summary tokens. The magnitude of improvement depends on the dataset: gains are modest when summary tokens suffice (e.g., CIFAR-100, EuroSAT), but substantial when spatial details are essential (e.g., GTSRB) or when the backbone distributes information across patch tokens, as in MAE. These results further support our claim that intermediate-layer fusion and patch-token selection operate on orthogonal

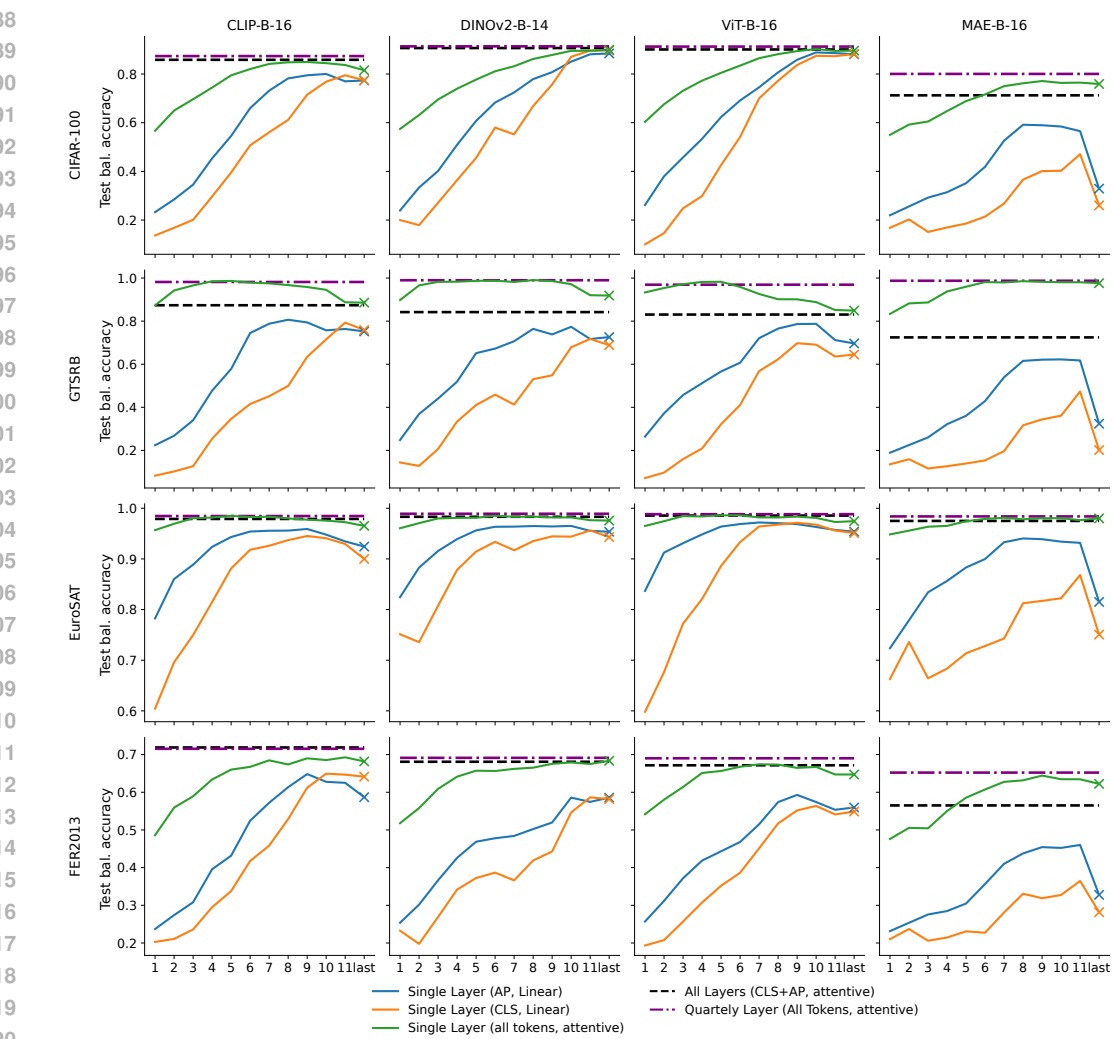

Figure 11: Downstream performance across intermediate layers for linear probe with `AP` and `CLS` token and attentive probe on all tokens. The dashed line indicates our multi-layer attentive fusion, which aggregates only the `CLS` and `AP` tokens across layers. Additionally, the purple dash-dotted line shows a hybrid approach, aggregating all tokens of intermediate layers.

representational axes and that incorporating intermediate-layer tokens is significantly more effective than relying solely on the last layer.

## A.9 PARAMETER EFFICIENCY COMPARISON

The linear probe and the attentive probe follow fundamentally different scaling behaviors. Given the hidden dimension $d$, the number of layers $|\mathcal{L}|$, and the number of classes $K$, the linear probe based on concatenation requires $2 \cdot |\mathcal{L}| \cdot d \cdot K + K$ parameters, scaling linearly with both the number of layers and the number of classes. In contrast, our attentive probe requires $8 \cdot d^2 + 10d + d \cdot K + K$ parameters, which scales quadratically with the embedding dimension $d$, linearly with the number of classes, and remains independent of the number of layers used. While the parameter count of the attention probe on all final-layer patches (AAT) is the same, its larger number of input tokens leads to higher computational costs.

Tab. 3 compares parameter counts across three Vision Transformer architectures over a range of class counts representative of the datasets in our experiments. While the attentive probe has a higher fixed overhead, its class-dependent growth is substantially slower than that of concatenation. As the

Table 3: Parameter counts for Linear and Attentive fusion probes using all layers (CLS+AP) across different numbers of classes ($K$) and three ViT architectures:
ViT-S ($d = 384$, $|\mathcal{L}_{\text{all}}| = 12$), ViT-B ($d = 768$, $|\mathcal{L}_{\text{all}}| = 12$), and ViT-L ($d = 1024$, $|\mathcal{L}_{\text{all}}| = 24$).

| | $d = 384, |\mathcal{L}_{\text{all}}| = 12$ | | $d = 768, |\mathcal{L}_{\text{all}}| = 12$ | | $d = 1024, |\mathcal{L}_{\text{all}}| = 24$ | |
| $K$ | Linear | Attentive | Linear | Attentive | Linear | Attentive |
|---|---|---|---|---|---|---|
| 2 | 18 434 | 1 184 258 | 36 866 | 4 727 810 | 98 306 | 8 400 898 |
| 5 | 46 085 | 1 185 413 | 92 165 | 4 730 117 | 245 765 | 8 403 973 |
| 10 | 92 170 | 1 187 338 | 184 330 | 4 733 962 | 491 530 | 8 409 098 |
| 50 | 460 850 | 1 202 738 | 921 650 | 4 764 722 | 2 457 650 | 8 450 098 |
| 100 | 921 700 | 1 221 988 | 1 843 300 | 4 803 172 | 4 915 300 | 8 501 348 |
| 200 | 1 843 400 | 1 260 488 | 3 686 600 | 4 880 072 | 9 830 600 | 8 603 848 |
| Backbone | 22 050 664 | | 86 567 656 | | 304 368 640 | |

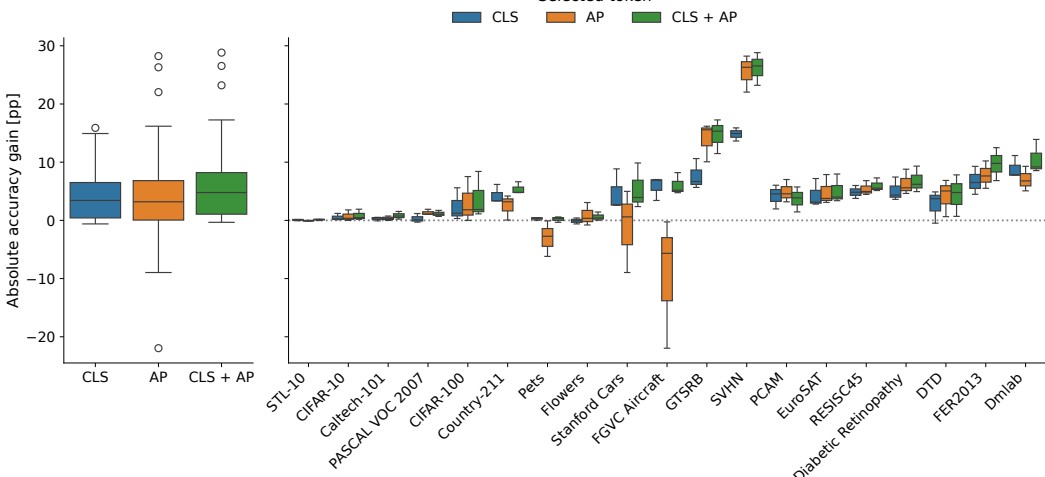

Figure 12: Absolute performance gains of attention-based intermediate layer fusion using different token configurations. Left: Distribution of gains across three base models and 20 datasets. Right: Per-dataset breakdown showing dataset-specific patterns in token utility.

class count increases, the linear probe grows rapidly, whereas the attentive probe remains relatively stable. In practice, the attentive probe uses fewer than 5% of the backbone's parameters, offering a highly efficient solution that scales well to large multi-class problems.

## A.10  IMPORTANCE OF INCLUDING STRUCTURAL INFORMATION

We analyze the effect of token selection in our attention-based intermediate layer fusion mechanism. We compare three configurations: attentive layer fusion using only CLS tokens, encoding the semantic information, only AP tokens, capturing more structural information by averaging spatial features, or both token types from all layers.

Fig. 12 shows absolute performance gains relative to the last layer CLS linear probe baseline across our three base models (CLIP-B-16, DINOv2-B-14, ViT-B-16) on all 20 datasets. We set the attention dropout to 0.1 to reduce the complexity of hyperparameter search.

The results demonstrate three key findings: (1) CLS tokens consistently provide positive gains across most datasets, (2) AP tokens exhibit high variance, substantially improving performance on some datasets (e.g., SVHN, GTSRB) while degrading it on others (e.g., FGVC Aircraft, Pets), and (3) combining both token types achieves the best overall performance, indicating the attention mechanism successfully learns when to utilize each token type.

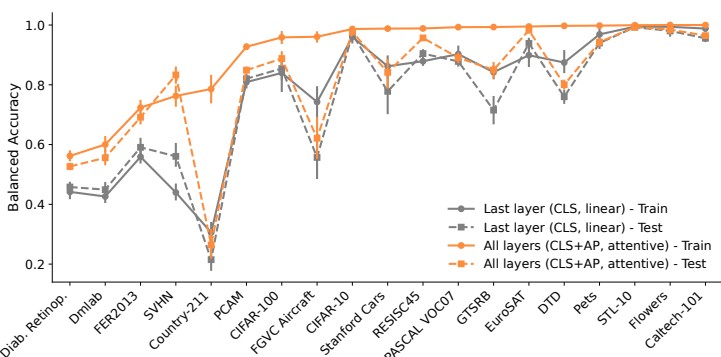

Figure 13: Train and test balanced accuracy comparison for each benchmark dataset across 9 models. The baseline performance (linear probe on last layer's `CLS` token) versus attentive probe on `CLS` +`AP` of all intermediate layers are shown. While most datasets show acceptable overfitting patterns, PCAM and PASCAL VOC 2007 exhibit overfitting where the attentive method's test performance approaches the linear baseline despite higher training accuracy.

## A.11 RISK OF OVERFITTING

More expressive probes inherently increase overfitting risk due to their greater capacity to memorize training-specific patterns. Despite mitigation strategies including weight decay and representational jittering, Fig. 13 reveals two overfitting patterns across our benchmark.

For most datasets, both methods exhibit similar train-test gaps, with our attentive fusion method maintaining superior test performance despite having a higher capacity. This represents acceptable overfitting where the additional expressiveness provides genuine benefits even with regularization. However, we observe overfitting on PCAM and PASCAL VOC 2007, where the linear baseline shows small train-test gaps while our attentive method overfits significantly despite regularization, resulting in test performance comparable to the simpler baseline (Tab. 1).

PCAM exemplifies this failure mode, potentially due to substantially more training updates (5,120 vs. 1,320 for our second-largest dataset) that may amplify overfitting effects. Additionally, standard data augmentation techniques could not be applied as regularization since we work with pre-extracted frozen features. Finally, the attentive fusion mechanism appears to overfit to noise in intermediate features, particularly from the `AP` token, which dilutes localized signals through spatial averaging, problematic since PCAM's diagnostic information concentrates in small tissue regions. By contrast, AAT avoids this issue despite a similar parameter count, as its attention mechanism operates only on the final layer and can thus focus directly on central patches. By contrast, AAT avoids this issue despite similar parameter count, as its attention mechanism operates only on the final layer.

This highlights a boundary condition: when label-relevant information is highly localized, `AP`-based aggregation becomes suboptimal, and limiting training steps becomes crucial even with regularization.

## A.12 IDENTIFYING THE OPTIMAL NUMBER OF HEADS

To determine the optimal number of attention heads for our approach, we conducted experiments using the DinoV2-B-16 model with all layers (CLS+AP, attentive pooling). While Chen et al. (2024) used 8 attention heads by default, we systematically evaluated different head configurations to identify the best setting for our method.

Due to computational constraints, we performed this analysis on a subset of 8 datasets: Stanford Cars, Country-211, GTSRB, CIFAR-100, DTD, EuroSAT, Pets, and SVHN. The experimental setup differed slightly from our main experiments by removing attention dropout, jitter, and gradient clipping to isolate the effect of the number of heads.

Fig. 14 shows that optimal performance is achieved when the number of attention heads equals the number of representations being fused, which we adopt for our method.

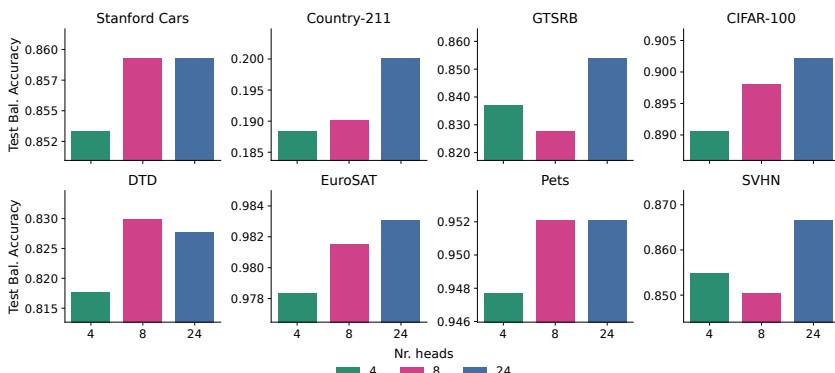

Figure 14: Test balanced accuracy across different numbers of attention heads on 8 datasets, showing optimal performance when heads equal representations fused.

## A.13 FINETUNING COMPARISON

Our work focuses on the probing paradigm, where the pretrained backbone remains frozen and only a lightweight classification head is trained. This approach is valuable in resource-constrained scenarios or when the model must serve multiple tasks and should therefore not be changed. However, to contextualize our contributions within the broader landscape of transfer learning methods, we conducted additional fine-tuning for the three base models (CLIP-B-16, DINOv2-B-14, and ViT-B-16) on GTSRB, CIFAR-100, and EuroSAT. Due to computational constraints, we used fixed hyperparameters for each model and dataset: learning rate $1 \times 10^{-3}$ and weight decay $1 \times 10^{-1}$ for the classification head, learning rate $1 \times 10^{-5}$ and weight decay $1 \times 10^{-6}$ for the backbone. We train for 40 epochs with a batch size of 256, enforcing at least 1000 gradient updates as in our main experiments.

Fig. 15 reveals that while fine-tuning generally achieves the highest accuracy, our method closely matches its performance on CIFAR-100 and EuroSAT. On GTSRB, fine-tuning achieves substantially higher accuracy (7-15pp), reflecting the value of direct backbone adaptation for fine-grained spatial discrimination. Importantly, our method consistently outperforms linear probing across all datasets while being 36 times faster during training compared to fine-tuning (Fig. 16), demonstrating a practical accuracy-efficiency trade-off for resource-constrained scenarios where probing is preferred.

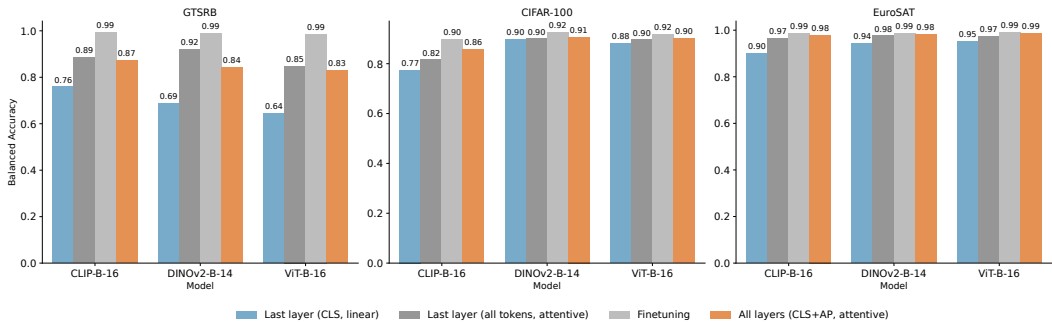

Figure 15: Downstream performance of three probing strategies and finetuning for three datasets (GTSRB, CIFAR-100, and EuroSAT) and the three base models.

## A.14 MULTI-LAYER FUSION ACROSS ATTENTION PROBE ARCHITECTURES

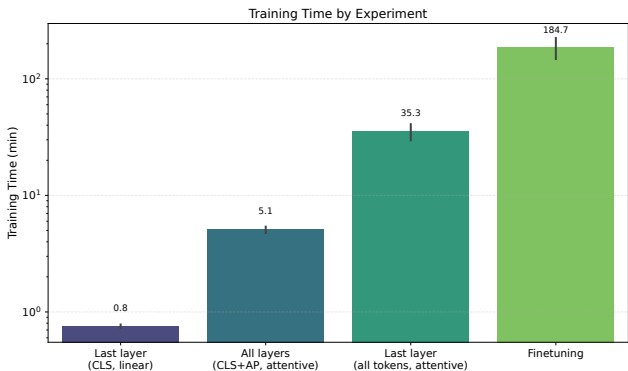

Figure 16: Training times in minutes for three probing strategies and finetuning averaged across datasets and the three base models.

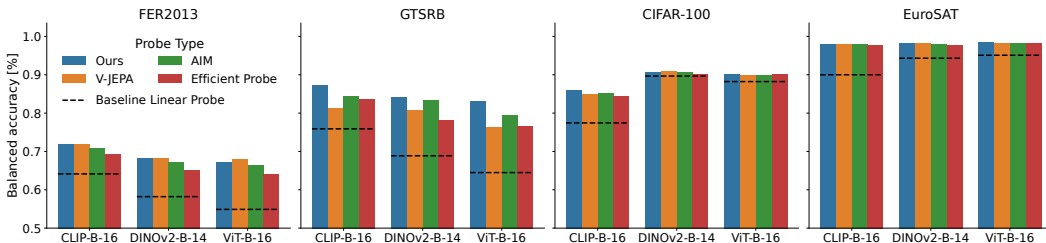

Figure 17: Performance of multi-layer attentive fusion using different per-layer attention probes. We compare our CAE-style probe with V-JEPA, AIM and Efficient Probe (EP), using both `CLS` and `AP` tokens from all layers.

To verify that the benefits of multi-layer attentive fusion do not depend on the specific design of the attention module, we evaluate several alternative attentive probes in place of our CAE-style implementation Chen et al. (2024). Specifically, we consider AIM El-Nouby et al. (2024), Efficient Probe (EP) Psomas et al. (2025), and V-JEPA Bardes et al. (2024), assessing their ability to aggregate intermediate layer information.

Fig. 17 reports accuracy on four representative datasets. All attentive probe variants consistently outperform the standard last-layer `CLS` linear probe, confirming that multi-layer attentive fusion effectively leverages intermediate representations independent of the attention design. Differences between probe types are minor, with more complex probes (CAE, V-JEPA) showing slightly higher gains on some tasks than the simpler variants (EP, AIM). In summary, the results confirm that multi-layer attentive fusion provides consistent downstream benefits across probe architectures, reinforcing the generality of our approach and validating the key claim that intermediate-layer features contain task-relevant information beyond the final layer `CLS` and `AP` tokens.

## A.15 STABILITY OF EXPERIMENT RUNS

To assess the stability of our experimental results, we conducted a seed variation analysis using the DinoV2-B-16 model with all layers (CLS+AP, attentive pooling). We ran five different random seeds for each of the 20 datasets in our evaluation. To reduce the hyperparameter search space, we removed attention dropout and focused the tuning process on learning rate and weight decay only. Fig. 18 shows the standard deviation of balanced test accuracy across the five seeds for each dataset. The results demonstrate that standard deviation remains below 0.01 for all datasets, with many datasets achieving standard deviations below 0.002. These values indicate that the variance across different random seeds is limited. Based on this stability analysis, we determined that single runs for each dataset and configuration would be sufficient for our main experiments, enabling us to allocate computational resources more efficiently while maintaining reliable results.

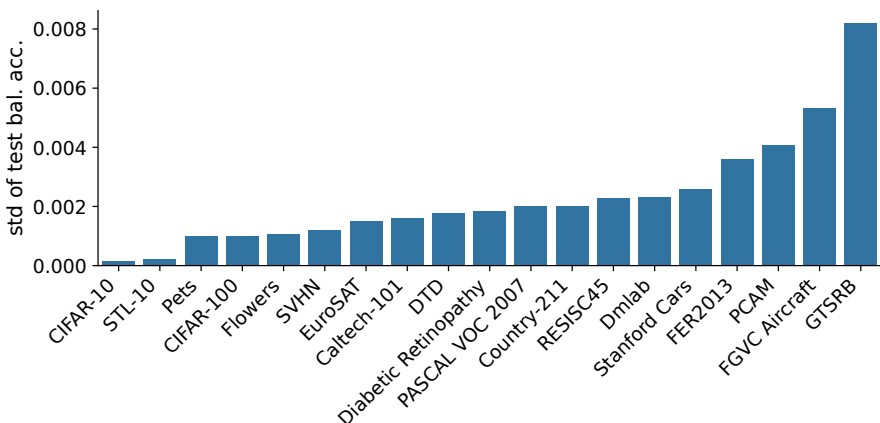

Figure 18: Standard deviation of balanced test accuracy across five random seeds for DinoV2-B-14 with all layers (CLS+AP, attentive pooling) on 20 datasets. All values remain below 0.01, indicating stable performance across different random initializations.

## A.16 USE OF LARGE LANGUAGE MODELS

Large language models (Google's Gemini, OpenAI's ChatGPT, and Anthropic's Claude) were used as a writing assistant to help refine the language and improve the clarity of the manuscript. Separately, AI-powered coding tools like Cursor and GitHub Copilot were used for advanced autocompletion during software development. The human authors directed all scientific aspects of the work, including the research ideas, methodology, and analysis of results, and are fully responsible for the content of the paper.

