# OpenReview forum: "Beyond the Final Layer: Attentive Multi-Layer Fusion for Vision Transformers"
_ICLR.cc/2026/Conference — Submitted to ICLR 2026_

### Official Review · Reviewer_FHUv · 2025-10-27

**Soundness:** 2
**Presentation:** 3
**Contribution:** 2
**Rating:** 4
**Confidence:** 4

**Summary:**

This paper studies probing-based adaptation, proposes a task-dependent attentive probing method on Vision Transformers (ViTs) to achieve better downstream task performance compared to the linear probing baseline. The key is to utilize intermediate layer representations (including CLS and average-pooled token embeddings), aiming at extracting complementary information therein.

**Strengths:**

- The paper is clearly written and is easy to follow.
- The experiment coverage is large, over 19 datasets and three model families.
- As claimed in the abstract and introduction, utilizing intermediate layer representation consistently outperforms the linear probing baseline by a substantial margin.
- The additional module scales only in the number of layers, not in the number of image patches, which remains reasonable even for high-resolution inputs and large base models.

**Weaknesses:**

- The main weakness is the practicality of the proposal. Though the method beats linear probing, its performance is on par with attentive probing on all last-layer tokens (AAT). No significant gain is observed when comparing with this baseline, and thus raises the question of why and when a user should switch to the proposal rather than probing use only the last layer token representations.

- This fact also questions the *motivation* of the proposal from the paper, where one would *expect benefits in some specialized domains, such as satellite imagery or medical images, due to the necessity of low-level structural cues contained in earlier layer representations*. From Table 1, (1) the benefits of the proposal are predominantly on tasks where “baseline accuracy is near saturation” (Line 362), while (2) on specialized / structured tasks, it does not always beat AAT. Both points contradict the motivation directly. With the additional result that on natural single domain tasks, the proposal usually performs worse than AAT, they together raise questions about the scope of the proposal on when it is practically useful, and further question the usefulness of intermediate layers.

**Questions:**

- In Section 4.2 (line 302), the proposal is described as having “markedly less variance” compared to AAT. However, from the figure, the var looks basically the same as AAT.

- In Section 4.3 (line 407), it is claimed that “the benefits are greatest for datasets outside the pretraining domain, where the CLS token alone often proves to be insufficient”. However, in the table, consistent gains are obtained on natural mult-domain tasks, which brings a direct contradiction.

---

> ### Author Response · Authors · 2025-11-21
> **Review Response**
>
> We thank the reviewer for their thoughtful assessment and recognition of our paper's clarity, extensive experimental coverage, and consistent improvements over linear probing. We address the main concerns below.
>
> ### **Main Concern: Practicality and Comparison with AAT**
>
> The reviewer asks when practitioners should choose our method over AAT. We argue that reliability across diverse settings is the key practical advantage.
>
> **Stability**: While AAT occasionally achieves higher peak performance, it exhibits substantially higher variance with frequent negative outliers (see Fig. 2). Our method delivers consistent gains across diverse tasks – critical when users cannot know a priori which method will work best. The empirical evidence for this can be seen in:
>
> - **Highest mean rank (1.45)** across all 20 datasets vs. AAT (2.74) (see Table 1)
> - **Positive gains over baseline on all 20/20 datasets**; achieves best or second-best performance on 18/20; AAT shows negative performance on 3 datasets
> - **Strong out-of-distribution performance** on specialized domains where baseline struggles: Diabetic Retinopathy (+6.86pp vs +1.94pp for AAT), RESISC45 (+5.23pp vs +4.07pp), EuroSAT (+4.37pp vs +3.38pp)
>
> **Practical Recommendation**: When facing new downstream tasks, particularly out-of-distribution applications (which are common real-world applications), our method provides a safer, more robust option than AAT's high-variance profile. Only when the task requires highly localised spatial information, AAT is the recommended choice, as our token aggregation looks at compressed information throughout the hierarchy. In addition, the training time for our method is 7x as fast as the training time for AAT.
>
> ### **Motivation and Domain-Specific Performance**
> The reviewer suggests a contradiction regarding specialized domains. We clarify:
> **"Near saturation" vs. specialized domains**: The reviewer conflates two distinct findings:
> - Natural multi-domain tasks (CIFAR-10, STL-10) show small gains because baseline accuracy is 96-99% (which is close to ceiling) – limited room for improvement
> - Specialized/structured tasks show large gains because they differ from the pretraining data, as we claimed.
>
> **Evidence supporting our motivation**:
> - Specialized (OOD): Diabetic Retinopathy (+6.86pp), RESISC45 (+5.23pp), EuroSAT (+4.37pp), PCAM (+2.85pp)
> - Structured: FER2013 (+10.05pp), Dmlab (+10.68pp)
> - Fine-grained natural: SVHN (+27.25pp), GTSRB (+13.47pp) - require structural cues beyond standard ImageNet pretraining
> - Fig. 4 confirms this: specialized datasets place substantially more attention on intermediate layers' AP tokens than natural multi-domain datasets.
>
> **Line 407**: "Benefits are greatest for datasets outside the pretraining domain" includes both specialized imagery AND fine-grained natural tasks requiring structural cues. Natural multi-domain tasks closest to the pretraining domain (e.g., CIFAR-10) show smallest gains, confirming our claim.
>
> **Efficiency**: As mentioned on page 4 (line ~200): "For ImageNet-sized input images, P ≈ 200, L ≈ 12, thus |L| << P yields an order of magnitude reduction in attention complexity." Our attention mechanism attends over just 24 summary tokens (12 layers × 2 tokens: CLS + AP), while AAT attends over ~200+ patch tokens. This makes our method:
> - Faster to compute (7x faster training time than AAT)
> - More memory efficient (number of layers is smaller than number of patch tokens)
> - Easier to scale to high-resolution images or larger models (with lots of patch tokens)
>
> ### **Specific Questions**
> **Q1: Variance (Line 302)**: Agreed - Fig. 2 shows AAT with wider spread and outliers. We will revise to state "AAT shows substantially wider spread with more frequent negative outliers."
>
> **Q2: Line 407**: There is no contradiction. We observe gains on all natural tasks, but largest gains occur on specialized/structured/fine-grained tasks that deviate more from pretraining, exactly as claimed. We will clarify this distinction.
>
> Our contributions can be summarized as follows:
> - **Reliability**: Best mean rank, positive gains on all 20/20 datasets
> - **OOD Performance**: Particularly strong on specialized or structured domains far from pretraining
> - **Safety**: No catastrophic failures unlike AAT
> - **Efficiency**: $\mathcal{O}\left(|L|^{2}\right)$ vs $\mathcal{O}\left(P^{2}\right)$ complexity (for the attention computation) where $|L|\ll P$ (number of layers is much smaller than the number of patch tokens)
> - **Interpretability**: Task-specific attention patterns (Fig. 4)
> For practitioners facing unknown tasks, particularly out-of-distribution applications, our method provides a principled, stable default with consistent improvements, whereas AAT's higher peak comes with substantial risk.

---

> > ### Comment · Reviewer_FHUv · 2025-11-24
> > **Comments on the rebuttal**
> >
> > Thanks to the authors for the rebuttal.
> >
> > *The usage of “variance” and stability*. I agree that the consistent positive gain over linear probing is a nice advantage of the proposal over the AAT baseline, but this property does not relate to “variance” in any sense. The phrasing is imprecise here since the discussion emphasizes no negative impacts rather than concerning the spread or quantile. According to figures 2 and 3, the spread and interquartile range are similar between AAT and the proposed fusion across all layers, and thus the original claim with respect to variance is not conceptually correct. A different phrase, like “stability” used in the rebuttal, might better fit the context to avoid confusion. Thanks to the authors for also responding to this in Q1 to improve the clarity of the paper.
> >
> > *The comparison with AAT, questions on motivation and contradiction*. I am fully aware that there are tasks near saturation, and thus the improvement can be marginal in these cases. However, I am not raising the concern regarding the absolute numbers in these cases or on specialized domains. The question is that, **compared to AAT**, the proposal does not offer absolute advantages other than stability. Regarding this comparison, according to Table 1, the only definitive conclusion one could draw is that the proposal beats AAT consistently on tasks near saturation (multi-domain tasks like STL-10, CIFAR-10, Caltech-101, and single-domain tasks like Pets and Flowers). On the rest natural image tasks and specialized/structured domains, the two methods outperform each other on a case-by-case basis. These are the reasons behind my concerns about the validity of motivation and the contradiction on benefits: they are not easy to argue or justify when AAT, instead of linear probing, is considered as the baseline.
> >
> > I appreciate the superiority compared to linear probing (as stated in the strength of the paper), and I think the stability advantage over AAT could be valuable. However, given the results in figures 2 / 3, and Table 1, it seems *when and how* the proposal could bring benefits in performance beyond stability over AAT is still unclear, as reflected in the weaknesses I spot.

---

> > > ### Author Response · Authors · 2025-11-27
> > >
> > > We thank the reviewer for the continued discussion and agree that our earlier use of the term variance was imprecise; we have revised the paper to refer to stability.
> > >
> > > Regarding the comparison to AAT, our intent was not to claim universal dominance, but to demonstrate that intermediate-layer fusion offers **complementary benefits**. AAT attends to high-resolution **spatial** detail in the final layer, whereas our method aggregates **hierarchical** information across all layers. These represent two orthogonal axes of representation, and therefore neither approach should be expected to consistently outperform the other across all tasks.
> > > Our results, however, show clear **benefits of hierarchical fusion**. Across the Natural Multi-Domain category, our method improves over AAT on all six datasets, including challenging and unsaturated ones such as:
> > > - Country-211 (+4.96pp vs −0.83pp for AAT)
> > > - CIFAR-100 (+3.33pp vs +1.73pp)
> > > - PASCAL VOC (+1.24pp vs −0.22pp).
> > >
> > > Similarly, on Specialized/Structured datasets, we outperform AAT on a clear majority (5/7), with the largest margins on domains farthest from the pretraining distribution, e.g., Diabetic Retinopathy (+6.86pp vs +1.94pp). The cases where AAT is stronger (GTSRB, PCAM, Dmlab) correspond to tasks that rely heavily on localized spatial cues, consistent with AAT’s patch-token design.
> > >
> > > To **further examine this orthogonality**, we added an experiment that **merges both approaches** on four datasets and base models by attending jointly over all tokens from layers {3, 6, 9} and the final layer. As shown in the updated **Fig. 11**, this combined variant consistently outperforms the original AAT baseline (green cross). Notably, on three datasets, the combined approach performs similarly to our summary-token method, suggesting that spatial patch tokens offer limited additional benefit when hierarchical cues dominate. In contrast, on GTSRB, where fine-grained spatial localization is crucial, the combined approach yields a substantial gain on all three base models (**98% accuracy for Intermediate Layer Fusion all tokens** vs 88.5% AAT vs 84.9% Intermediate Layer Fusion vs 69.7% Linear CLS), demonstrating that, also here, **incorporating intermediate-layer tokens is significantly more effective than relying solely on the last layer**.
> > >
> > > Together, these analyses clarify the scope highlighted by the reviewer: our method is most beneficial when tasks depend primarily on hierarchical or global cues, while AAT is preferable for tasks requiring high-resolution spatial detail. The two approaches are therefore complementary rather than competing, and the additional combined experiment confirms their orthogonality in practice. We emphasize these two axes in our updated introduction and discussion.

---

### Official Review · Reviewer_XzCg · 2025-10-31

**Soundness:** 2
**Presentation:** 3
**Contribution:** 3
**Rating:** 4
**Confidence:** 5

**Summary:**

The paper tackles the question: “Can we do better than probing only the last layer of a frozen ViT?” The authors argue that task-relevant information is distributed across the depth of a ViT and not fully captured by the final CLS token. They therefore propose an attentive multi-layer fusion probe that, for each downstream task, attends over CLS and average-pooled (AP) tokens extracted from all transformer layers and learns to weight them via a lightweight cross-attention module. This keeps the backbone frozen and trains only the probe. Evaluated on 19 diverse datasets and three model families (CLIP, DINOv2, supervised ViTs), the method consistently improves over standard last-layer linear probing and is more stable than attentive pooling over all last-layer tokens (CLS + AP). The analysis of the learned attention maps shows that tasks close to pre-training focus on later layers, whereas domain-shifted/specialized tasks draw more from intermediate AP tokens, supporting the paper’s main claim about the usefulness of intermediate representations.

**Strengths:**

1. **Timely and relevant problem.** Attentive probing is becoming increasingly important as full/PEFT fine-tuning gets more expensive and also changes the backbone; probing instead tells us what the pre-trained model already knows. Positioning the paper as “probing as evaluation of representational potential” is spot on.
2. **Clear, well-written, easy to follow.** The method is essentially “extend attentive probing from one layer to all layers, over CLS+AP,” but the authors explain it cleanly, with a good schematic (Fig. 1) and a nice notation for layer subsets (last, mid+last, quarterly, all).
3. **Broad experimental sweep.** 19 datasets, three major pre-training paradigms (CLIP / DINOv2 / supervised), and small–base–large variants. This gives the reader confidence that the effect is not cherry-picked.
4. **Good reproducibility.** Code + detailed appendix are promised and the paper already contains enough detail to reimplement the probe.
5. **Insightful analysis sections (Sec. 4.4–4.5)**. The heatmap analysis that shows when intermediate layers are attended to (domain shift, structural/satellite/medical tasks) and what token types matter (AP used across more layers than CLS) is genuinely informative — I have not seen this exact “which layers matter for which kind of shift” angle before. This is a real contribution to our understanding of probing, not just to accuracy

**Weaknesses:**

Major:

1. **Not the first multi-layer attentive probing**. The paper presents the idea as if “attentive probing = last layer” and “we are the first to go across layers.” But there is earlier work (e.g. Psomas et al., 2025) that already applies attentive probing independently to multiple layers of an encoder and reports that attentive probes beat linear probes at basically every depth. What seems novel here is specifically the joint, cross-attentive fusion of CLS+AP across all layers. That narrower novelty should be stated more explicitly.
2. **Gains over the strongest attentive baseline are small/inconsistent**. The central technical claim is “attend over all layers > attend over last layer.” But looking at the exact numbers (Figure 5): for CLIP models the mean gain is ≈1 pp, for DINOv2 it can even go slightly down, and for supervised ViTs it is roughly on par. This is not the kind of margin that kills alternative designs — it says “this is a nice refinement,” not “this is the new default.” So the paper somewhat *overstates the strength of the result*.
3. **Single attentive baseline (CAE - Chen et al., 2024)**. The whole paper is built around one attentive pooling choice (the CAE-style cross attention). Yet the attentive-probing literature has already several flavours (AIM [1], EP (Psomas et al., 2025) / multi-query attentive pooling, heavy V-JEPA-style [2] pooling). Since the core contribution here is “multi-layer attentive fusion,” the natural question is: does this generalise to other attentive poolings? If yes, show 1–2 of them; if not, at least discuss why. Right now the reader can wonder whether the results are a quirk of CAE.
4. **AIM and CAE are very close**. AIM [1] and the employed CAE-style probing both rely on a learned query and effectively absorb W_Q. The paper could likely drop W_Q and simplify the probe, or at least comment on this — especially since one of the selling points of probing is being lightweight.
5. **Missing experiment that would strengthen the story**. A simple table/plot with “independent linear probe per layer, CLS vs AP,” followed by “independent CAE probe per layer” would vividly show why we need this layer fusion and whether the proposed layer-fusion performance is an upper bound when compared with independently linear (CLS or AP) or attentive (CAE) probing performance.
6. **Backbone diversity could strengthen the story.** All results are on CLIP / DINOv2 / supervised ViT. But masked-image-modeling or generative models (e.g. DiT) are the types of models probing papers ([3]) report large attentive-probe gains on — adding one of these could give the paper the extra empirical “punch” it currently lacks.

[1] El-Nouby, Alaaeldin, et al. "Scalable pre-training of large autoregressive image models." arXiv preprint arXiv:2401.08541 (2024).
[2] Bardes, Adrien, et al. "Revisiting feature prediction for learning visual representations from video." arXiv preprint arXiv:2404.08471 (2024).
[3] Przewięźlikowski, Marcin, et al. "Beyond [cls]: Exploring the true potential of Masked Image Modeling representations." Proceedings of the IEEE/CVF International Conference on Computer Vision. 2025.

Minor:
1. Metric choice obscures the effect. Most plots are in “absolute accuracy gain over last-layer CLS linear probe.” This is fine for the headline (“probing deeper helps”), but it becomes confusing once we want to compare attentive-vs-attentive or last-layer attentive vs multi-layer attentive.
2. GAP (global average pooling) might be better/more commonly used than AP.

**Questions:**

1. Related to weakness 3: Can you run the exact same multi-layer fusion but with EP (Psomas et al., 2025) or AIM [1] as the per-layer attentive block to show that the gains are not CAE-specific?
2. Related to weakness 4: In your setting, could you remove W_Q (as AIM effectively does) without hurting performance? If not, why is your query different?
3. Related to weakness 5: Could you add this experiment + discussion about it.
4. Related to weakness 6: Could you add a MIM backbone (e.g. MAE or CAPI)?

---

> ### Author Response · Authors · 2025-11-21
> **Response to Weaknesses**
>
> We thank the reviewer for the thoughtful and constructive feedback. The suggestions helped us significantly strengthen the paper, and we address each concern point-by-point below.
>
> ### **Major Weakness 1:**
> We agree that we are not the first to apply attentive probes to intermediate layers, but our contribution is orthogonal: prior work evaluates layers independently, while we introduce a joint cross-layer fusion mechanism that aggregates information across the network hierarchy, operating through summary tokens (CLS/AP). We argue that this application of attentive probes for hierarchical aggregation is new and empirically valuable.
>
> ### **Major Weakness 2:**
> We agree that improvements are not large in all settings. We updated the manuscript (Discussion) to make explicit that intermediate-layer fusion is most effective when summary tokens capture task-relevant information and downstream tasks do not require fine spatial details, a finding extended through your suggested MAE experiments (clf. Answer to Q4). Under these conditions, our method yields consistent gains across models and datasets, while being more efficient than attending over all tokens  ($L << P$).
>
> ### **Major Weaknesses 3-6:**
> See responses to questions 1-4.
>
> ### **Minor Weakness 1:**
> We agree that plotting relative accuracy can be confusing. However, because performance varies substantially across datasets and models (see left panel of Fig. 3), showing improvements relative to the standard linear probe is the clearest way to highlight differences when aggregating over models or datasets.
>
> ### **Minor Weakness 2:**
> Our AP token corresponds to global average pooling, as defined in Equation 1. While global average pooling is a more precise name, we chose AP for conciseness as the full spatial aggregation becomes clear from the context.

---

> ### Author Response · Authors · 2025-11-21
> **Response to Questions**
>
> ### **Q1 Alternative attentive probe architectures**
> Following the request, we evaluated three alternative attentive probes, AIM (El-Nouby et al., 2024), Efficient Probe (EP; Psomas et al., 2025), and V-JEPA (Bardes et al., 2024), within our multi-layer fusion framework. The new results in **Appendix A.14 (Fig. A.17)** show:
> - All attentive variants consistently outperform the last-layer linear probe across four representative datasets and three base models (CLIP-B/16, DINOv2-B/14, ViT-B/16).
> - Performance differences between probe types are minor.
> - The more expressive probes (CAE, V-JEPA) achieve slightly higher absolute gains, but every attentive design effectively leverages intermediate representations.
>
> These results confirm that the benefit arises from multi-layer fusion itself, not from the CAE-specific architecture.
>
> ### **Q2 Removing $W_Q$ and relation to AIM**
> Our attentive probe differs from AIM in several architectural details beyond the presence of $W_q$:
> - projection of the query, key, and value vector to a higher-dimensional space ($d \to 2d$)
> - LayerNorm applied to the query token for amplitude regularization
> - a post-attention linear-BatchNorm-linear head (AIM uses a single linear layer)
> - attention dropout for regularization.
>
> These components improve training stability when attending across multiple intermediate layers. Removing  $W_Q$ (as in AIM) does not significantly degrade performance in preliminary tests (average diff of 0.08 pp across 4 datasets). However, we retain the original CAE form in the main paper for consistency.
>
> ### **Q3 Per-layer linear and attentive probe**
> Following the reviewer’s request, we have now reorganized and expanded our original Appendix Section 6 into two dedicated sections. Appendix A7 continues to discuss the relationship between representational similarity and linear-probe performance, while the new **Appendix A8 (with Fig. 11)** directly compares the proposed per-layer strategies: linear probing on the CLS and AP tokens and an attentive (cae) per-layer probe that aggregates all spatial tokens. These results show that:
> - Attentive fusion consistently outperforms its individual components and substantially exceeds any single-layer probe on CLS or AP tokens.
> - For GTSRB, highly localized features are crucial, leading to better performance of attentive probes across all tokens.
> - Attention over all tokens often performs best in intermediate layers, highlighting the value of capturing information distributed across the network hierarchy.
> - For MAEs, meaningful information is spread across patch tokens, causing AP to discard structures that an all-token attentive probe can still exploit.
>
> Overall, these results confirm that our fusion recovers complementary information distributed across depth, while intentionally restricting itself to CLS/AP–based aggregation for efficiency and interoperability.
>
> ### **Q4 Adding a Masked Image Modeling backbone**
> We have added experiments on MAE (both base and large) and report the results in the new **Appendix Section A6**:
> - Relative to a linear probe on the AP token (the appropriate summary token for MAE, since its CLS token is unused during pretraining), our intermediate-layer attentive fusion yields substantial gains of 22.4 / 24.7 percentage points on average (**Appendix Fig. 8**). This confirms that multi-layer aggregation also provides large improvements for masked-image-modeling backbones (**Appendix Fig. 7**).
> - MAE exhibits a fundamentally different representational structure than CLIP, DINOv2, or supervised ViT. Because MAE does not learn a CLS token and distributes information across patch tokens, full-token attention is naturally advantaged for this model family.
> - Our attention maps (**Appendix Fig. 9**) confirm this behavior: the attentive probe assigns negligible weight to the CLS token and consistently focuses on AP-derived information across layers.
>
> These findings align with prior observations that MIM models retain fine-grained, spatially localized information until late layers and therefore benefit most from probes that can attend to all tokens. **This reflects a limitation of the token-aggregation interface, not of intermediate-layer fusion itself**.
> Importantly, token-level aggregation (CLS/AP vs. all tokens) and layer-level aggregation (single-layer vs. multi-layer) address orthogonal questions:
> - When the model compresses information into summary tokens (CLIP, DINOv2, supervised ViT), fusing layers over these tokens is highly effective.
> - When the model does not (MAE), probes that access all tokens are naturally stronger. Yet, layer selection remains an open question even when attending to all tokens.
>
> Thus, the MAE results reinforce our central claim: **hierarchical layer fusion provides substantial gains over its last-layer counterpart**, and is especially effective when the model provides meaningful layer-wise summary representations.

---

> ### Comment · Reviewer_XzCg · 2025-11-26
> **Updated Review**
>
> I would like to thank the authors for this *great rebuttal*. I would like to focus on:
>
> - The fact that indeed prior work mostly evaluates layers independently, while the authors introduce joint cross-layer fusion mechanism. There is some novelty coming from *"independently"* vs. *"jointly"*.
> - The MAE results provide some useful insights for the community. The authors mention these clearly. When the model compresses (most of the) information into summary tokens (CLIP, DINOv2, supervised ViT), fusing layers over these tokens is *highly effective*. When the model does not (e.g., MAE), *probes that access all tokens are naturally stronger*.
>
> Motivated by the novelty and these insights, I increase my rating and lean towards accepting. However, I cannot oversee the fact that improvements are marginal in some settings.

---

> > ### Comment · Reviewer_XzCg · 2025-11-26
> > **Follow-up Question**
> >
> > I would also like to ask two more questions.
> >
> > 1. Is there any chance to try (as an ablation experiment), an alternative formulation of your mechanism, in which (only for models that distribute information across patch tokens - like MAE) you use the internal MHSA for top-K (maybe top-2 to be consistent with what you already do) token selection per layer? The idea is that in these models indeed the information is spread to patch tokens and indeed there is not a "summary" token. Using [CLS] or AP is not working that well. But what if you could somehow exploit the internal MHSA to choose top-K patch tokens per layer? Would this give better results?
> >
> > 2. In implementation details, I just saw that for the AAT baseline you use 8 attention heads. I have a feeling that if you increase the number of heads to 12, 16, 32, this baseline is going to become even stronger. Could you provide a small analysis/ablation about this too?
> >
> > Thanks a lot, once more!

---

> > > ### Author Response · Authors · 2025-11-27
> > > **Response to Follow-Up Questions**
> > >
> > > We thank the reviewer for the positive assessment of our rebuttal and for raising two thoughtful follow-up questions. We address both below.
> > >
> > > ### **Q1 Combining patch token from intermediate layers**:
> > > We agree this is an interesting direction, and we added an experiment aimed at clarifying the orthogonality between our intermediate-layer fusion and patch-token aggregation methods such as AAT. While designing a principled strategy for constructing advanced summary tokens from patch tokens would require careful investigation and, in our view, merits a dedicated study of its own, we included a **naïve hybrid experiment** that combines both hierarchical and spatial information to approximate the reviewer’s suggestion.
> > > Specifically, we fuse **all tokens** (patch + CLS) from three **quarterly spaced intermediate layers** (3, 6, 9) together with the final-layer tokens, and apply our attention probe over this enlarged token set. This hybrid approach leverages **both hierarchical information** (from intermediate layers) and **spatial detail** (from patch tokens). As shown in the updated **Fig. 11**, it consistently outperforms both AAT applied to the last layer, as well as our approach relying only on summary tokens. The magnitude of improvement depends on the dataset: gains are small when summary tokens suffice (e.g., CIFAR-100, EuroSAT), but substantial when spatial details are essential (e.g., GTSRB) or when the backbone distributes information across patch tokens, as in MAE.
> > > These results further support our claim that intermediate-layer fusion and patch-token selection operate on orthogonal representational axes.
> > >
> > > ### **Q2 Sensitivity of the AAT baseline to the number of attention heads**:
> > > We used the same configuration as the CAE paper to ensure strict comparability with prior work. Following the reviewer’s suggestion, we re-ran the AAT baseline on the three base models (DINOv2-B/14, Clip-B/16, ViT-B/16) and four representative datasets (FER2013, GTSRB, CIFAR-100, EuroSAT) using the suggested 12, 16, and 32 heads. The average change in accuracy relative to 8 heads is:
> > > - 12 heads: +0.33 pp (std 0.26)
> > > - 16 heads: +0.11 pp (std 0.51)
> > > - 32 heads: +0.61 pp (std 0.63)
> > >
> > > While these results indicate that 8 heads is not necessarily optimal, the differences are very small (<0.7 pp), while more head leads to decreased stability across models. This confirms that the improvements we report are based on the use of intermediate-layer fusion rather than specific AAT hyperparameter choices. We have also added a brief note in the implementation section of the manuscript reflecting this result.

---

### Official Review · Reviewer_5xNU · 2025-11-01

**Soundness:** 2
**Presentation:** 2
**Contribution:** 2
**Rating:** 2
**Confidence:** 4

**Summary:**

This paper studies the problem of transferring large-scale vision foundation models to downstreams tasks. The authors find that linear probing is restricted to last-layer representations which limits the performance as task-relevant information is distributed across the network hierarchy rather than solely encoded in any of the last layers. To address the issue, the authors propose an attentive probing mechanism that learns to identify the most relevant layers for a target task and combines low-level structural cues with high-level semantic abstractions to dynamically fuses representations from all layers of a Vision Transformer. Empirical results on multiple datasets and pretrained foundation models demonstrate the effectiveness of the proposed method.

**Strengths:**

1. The experiments are extensive. It is appreciated that the authors have conduct experiments on different foundation models and various fine-grained classification datasets.
2. The writing is clear and the paper is easy to follow. The overall structure is well organized and the idea is presented in a coherent manner.

**Weaknesses:**

1. The baseline of the paper is inappropriate. The authors seems to only compare the proposed method with linear probing, which is problematic as linear probing itself is more like an evaluation protocol to test the generalization ability of foundation models. In other words, linear probing can hardly be regarded as a transfer learning method which aims to persuit state-of-the-art performance. In the transfer learning context, better baselines could be full fine-tuning and efficient fine-tuning methods like LoRA.
2. The proposed method seems heuristic as the authors seem to leverage different settings for different experiments which means the proposed method does not have a general solution which could work for different test cases. It is suggested to develop the method in a systematic way.
3. While the authors have provided abundant results on fine-grained classification, additional results on ImageNet-1K experiments and dense prediction tasks like semantic segmentation would help to better understand the behavior of the proposed method.

**Questions:**

see weakness above

---

> ### Author Response · Authors · 2025-11-21
> **Review Response**
>
> We thank the reviewer for the feedback and address each weakness below.
>
> ### **Weakness 1: Missing fine-tuning baselines for transfer learning**
> We respectfully note that our **paper specifically targets the probing paradigm**, where the backbone remains frozen and only a lightweight head is trained. This paradigm is valuable when: (1) computational resources are limited, (2) the model must remain general-purpose for multiple tasks, or (3) backbone modification is restricted. Within this paradigm, we compare against both standard **linear probing** and the **attentive probing** method (AAT; Chen et al. 2024), which attends to all tokens in the final layer.
>
> Nevertheless, we contextualize our contributions and have **added fine-tuning experiments in a new Appendix section A.13** to address it.
>
> **Experimental Setup**:
> - three datasets (CIFAR-100, GTSRB, EuroSAT)
> - all three base models (CLIP-B-16, DINOv2-B-14, ViT-B-16)
>
> **Results**: While fine-tuning achieves highest accuracy on some datasets, our method matches it on others (e.g., CIFAR-100, EuroSAT) and always outperforms linear probing, while being 33× faster to train (**Appendix Fig. 15-16**).
> This demonstrates a practical accuracy-efficiency trade-off for resource-constrained scenarios.
>
> ### **Weakness 2: The method appears to be a heuristic**
> We respectfully disagree with this characterization. Our method uses a single, systematic design across all experiments:
> - Layer selection: All layers (L_all) for every dataset and model
> - Token types: Both CLS and AP tokens throughout
> - Fusion mechanism: Multi-head cross-attention with heads = 2×|L|
> - Architecture: Same attention-based fusion (Equations 3-5)
>
> **What varies**: Only hyperparameters (learning rate, dropout, weight decay), selected via standard grid search on a validation set. This is common practice for any probing method and does not make the approach heuristic. Note that we perform the same hyperparameter search for all methods as detailed in **Appendix A.1**.
>
> **Ablation studies**: Our experiments testing different layer subsets (L_last, L_mid+last, L_quarterly) or token types (CLS only, AP only) are systematic ablations to validate design choices, not different configurations of our method. They demonstrate why our choices (all layers, both tokens) work best.
>
> The method itself is a general solution that automatically adapts to each task through learned attention weights—this adaptive layer-fusion behavior is the core contribution. We would appreciate clarification on which specific "different settings" the reviewer observed, as this would help us improve our presentation.
>
> ### **Weakness 3: Missing ImageNet-1K experiments and dense prediction tasks (semantic segmentation)**
>
> We appreciate this suggestion and have **added ImageNet-1K results to Table 1**. Due to the dataset's size, AAT experiments are still running and will be added upon completion, **we will notify you when available.**
>
> **Regarding dense prediction tasks**: Our work focuses specifically on image classification, which allows controlled evaluation across 20 diverse datasets spanning multiple domains (natural, specialized, structured) and 9 model variants, where the frozen-backbone probing paradigm is most established. While the core insight, that intermediate layers contain task-relevant information, is well-established in dense prediction architectures (e.g., U-Net, FPN), adapting our attention-based fusion mechanism to segmentation or detection would require architectural modifications to handle spatial outputs rather than global predictions. We consider this valuable future work but beyond the current scope, as our contribution focuses on demonstrating the value of adaptive intermediate-layer fusion within the classification probing paradigm.

---

> > ### Author Response · Authors · 2025-11-27
> > **Additional ImageNet results**
> >
> > Dear Reviewer,
> >
> > We are writing to inform you that all ImageNet runs have successfully completed.
> > We have updated the manuscript to include these final results, specifically in **Table 1** and all main figures (**Fig. 2 and Fig. 3**), as well as the detailed model performances in the appendix (**Fig. 5 and Fig. 8**).
> >
> > We appreciate you taking the time to review our rebuttal and look forward to your assessment.

---

### Official Review · Reviewer_MGWs · 2025-11-01

**Soundness:** 3
**Presentation:** 4
**Contribution:** 3
**Rating:** 6
**Confidence:** 4

**Summary:**

This paper proposes a novel method to aggregate representative tokens (like [CLS] from multiple layers. The authors claim that the intermediate tokens would have rich information relevant to downstream tasks based on the previous researches. The authors propose an attentive probing method and represent performance gain on various tasks.

**Strengths:**

1. The evaluation is broad and through.
2. Incorporating AP token is interesting.
3. The attentive mechanism shows meaningful improvement over linear model.

**Weaknesses:**

1. The model size (parameters of attentive probing) is huge compared to linear model. Also, it will increase quadratically  with base model's depth.
2. For fine graned tasks, all token from the last layer performs better.

**Questions:**

1. How can one find whether  attentive probing performs better than all tokens method? Is applying and evaluation the only method for real applications?
2. How to overcome the potential overfitting problem?
3. Not in cases, the overhead of attentive probing over linear model is not justifiable. How can one decide which method would be better fit considering the overhead and risk of overfitting?

---

> ### Author Response · Authors · 2025-11-21
> **Response to Weaknesses**
>
> We thank the reviewer for the thoughtful and constructive feedback, and for recognizing our broad evaluation and meaningful improvements. Below, we address the important questions raised about model complexity, robustness, and probe selection.
>
> ### **Weakness 1: Model Size, Parameter Scaling, and Efficiency**
>
> **Clarification: Parameter count does not grow quadratically with depth.**
> We would like to clarify that the parameter count of our attentive probing module is independent of the number of layers $L$ and does not scale quadratically. As shown in Table 3, the dominant factor for parameter size is therefore the hidden dimension $d$, not the depth of the network. **See changes in Section 3.1**.
>
> **Efficiency/computational complexity advantage: $\mathcal{O}(|L|^2)$ vs. $\mathcal{O}(P^2)$.**
> Our probe attends over 2|L| summary tokens, leading to attention complexity $\mathcal{O}(|L|^2)$.
> AAT-style methods attend over all P patch tokens in the final layer, with complexity $\mathcal{O}(P^2)$ where:
> - $P \approx 200$ (ImageNet resolution)
> - $|L| \approx 12$
>
> Thus, $|L| << P$ and our method is **$\approx 7x$ more efficient** in practice while leveraging information from all layers (**Appendix Fig. 16**). This was a key motivation behind our design: it keeps the probe efficient, stable, and easy to scale to larger backbones or higher-resolution inputs.
>
>
> ### **Weakness 2: Overall Performance and Robustness Across Tasks**
>
> **Clarification: Our method is consistently strong, even when not the top performer in few cases.**
> It is correct that AAT exceeds our probe on a _few highly fine-grained_ tasks. However:
> - Our method is the **second-best** in nearly all of these cases.
> - Importantly, across all 20 datasets, we achieve the **best average performance and the best mean rank** (**Tab. 1**).
> - We obtain positive gains on every dataset, demonstrating consistent improvements rather than task-specific specialization.
>
> **Why this matters for robustness.**
> AAT’s peak performance comes at the cost of higher variance and greater overfitting risk, as shown in **Fig. 2** (larger spread, frequent negative outliers). Our approach provides:
> - **Lower variance**,
> - **Stable improvements across domains**, and
> - **No catastrophic failures**,
>
> making it a safer and more reliable option for practitioners, particularly when task characteristics are unknown.
>
> **Manuscript updates.**
> We have extended the discussion in **Section 4.4** to highlight this robustness and further clarify why our method offers the best overall trade-off across diverse tasks. We also refer the reviewer to our general response, where this point is elaborated in more detail.

---

> > ### Author Response · Authors · 2025-11-21
> > **Response to Questions**
> >
> > ### **Q1: When does attentive probing outperform attending to all tokens?**
> > **General trend: our method is the more reliable default**
> > Across the 20 datasets, we observe that attending over all intermediate layers provides the most consistent improvements.
> > Tasks that are **outside the pretraining domain** or require **structural or mid-level reasoning** (e.g., EuroSAT, FER2013, DMLab) benefit from intermediate-layer information, which final-layer patch embeddings tend to abstract away. Intermediate layers preserve broader structural cues, leading to more stable gains.
> >
> > **When AAT can be preferable**
> > AAT occasionally outperforms our method on a small set of **highly fine-grained natural single-domain tasks** (Stanford Cars, FGVC Aircraft, GTSRB), where extremely subtle spatial details are crucial. In these cases, direct access to all patch tokens can yield small additional improvements because the final-layer patch embeddings explicitly capture fine-grained spatial variation.
> >
> > **Practical conclusion**
> > For the vast majority of tasks, and especially when task properties are unknown, **attending over intermediate layers generalizes best**, offering a stable and robust choice.
> > AAT is only occasionally optimal for extremely fine-grained tasks or when the backbone does not use a summary token during training.
> >
> > **Manuscript update:** We have added a short explanation of this intuition in **Section 4.4**.
> >
> > ### **Q2: How do you prevent overfitting in the attentive probe?**
> > **Regularization strategy (Appendix A.9)**
> > To mitigate overfitting, we apply several standard but effective techniques:
> > - Weight decay on all probe parameters
> > - Dropout inside the attention module
> > - Jittering of intermediate-layer representations
> > - Hyperparameter tuning based on a validation split
> > - Class-balanced loss to avoid bias toward majority classes
> >
> > Together, these reduce sensitivity to spurious correlations or individual layers.
> >
> > **Lower inherent overfitting risk compared to AAT**
> > Our probe attends only to **summary tokens (CLS + AP)** which already represent compact representations. In contrast, AAT attends to **hundreds of patch tokens**, substantially increasing dimensionality and overfitting risk. This difference is reflected in our results: **our method exhibits a considerably lower variance** across datasets (Fig. 2 & Fig. 3).
> >
> > **Manuscript update:** We have updated **Section 4.1** to explicitly describe these regularization measures for improved transparency.
> >
> > ### **Q3: When is the overhead of attentive probing justified compared to a linear probe?**
> > **Consistent gains justify the overhead**
> > Across all datasets (**Tab. 1**), our attentive probe delivers positive and substantial improvements over the linear probe. Thus, whenever downstream performance or robustness across diverse tasks matters, attentive probing provides a clear advantage.
> >
> > **Predictable and modest computational cost**
> > As described in Sec. 3.1, the probe's parameter count scales primarily with the hidden dimension d. Its cost is therefore generally modest. Importantly, the probe operates on summary tokens only, not full patch sequences, further reducing overhead.
> >
> > **Training time differences are small**
> > Training time differences versus linear probing are minor (**Appendix Fig. 16**) while accuracy gains remain substantial. Additionally, training time of multi-layer fusion is 7x faster than attending to all tokens of a layer. This makes multi-layer fusion the preferable choice unless extreme parameter minimization is required or the task is exceedingly simple.
> >
> > **Manuscript updates:**  We have revised **Section 4.4** and **Section 5** to clarify when the overhead of attentive probing is justified and how it compares to linear probing in practice.

---

### Author Response · Authors · 2025-11-21
**General Response**

We thank all reviewers for their careful reading of the paper and for many constructive comments. The reviews unanimously highlight three key strengths:
- **Relevance and Contribution:** Reviewers recognized the work as "timely and relevant" (XzCg) and found the incorporation of AP tokens "interesting" (MGWs), noting that the method yields "meaningful improvement" (MGWs) and "consistently outperforms the linear probing baseline" (FHUv).
- **Thorough Evaluation:** Reviewers commended the experimental rigor, describing it as "broad and thorough" (MGWs), "extensive" (5xNU), and a "broad experimental sweep" (XzCg) that covers "large experiment coverage" (FHUv).
- **Clarity:** The manuscript was praised as "clear, well-written" (XzCg), "easy to follow" (FHUv), and presented in a "coherent manner" (5xNU).

We are grateful for this consensus. We now address the shared observation regarding the All-Tokens (AAT) baseline to clarify our method’s specific scope and utility. We agree that AAT (attending to all tokens in the last layer) may outperform our multi-layer probe on specific fine-grained datasets, but our goal is not a specialized solver for fine-grained recognition. Instead, we propose an **adaptive probing strategy that exploits intermediate features** for robustness across unknown downstream tasks. Importantly, our added experiments during the rebuttal revealed that hierarchical fusion (across layers) and spatial fusion (across patch tokens) address orthogonal representational axes, which can be combined for further improvements.

Given these insights, we argue that our Multi-Layer Fusion represents a highly valuable contribution as an advanced probing strategy for diverse downstream tasks:
- **Best Mean Rank (1.45 vs. 2.74)**: Across all 20 datasets and 9 models, our probe achieves the best mean rank (1.45) among all methods. In contrast, AAT ranks substantially lower (2.74).
- **Universal Improvement (20/20)**: Our method yields gains over the standard linear probe on every dataset. Conversely, AAT exhibits high instability: while it peaks on texture-heavy tasks, it suffers from negative transfer on others (e.g., Country-211, PASCAL VOC), performing worse than a simple linear probe.
- **Interpretable Attention Heatmaps**: The learned query token shows which layers each task relies on, giving a clear view of how tasks tap into the model’s representational hierarchy.
- **Computational Complexity**: Our method scales with network depth ($\mathcal{O}(|L|^2)$), while AAT scales with patch tokens ($\mathcal{O}(P^2)$). Since $|L| \ll P$ (typically $|L|=12$ vs. $P \approx 200$), our approach is order of magnitude more efficient, making it a more feasible attention mechanism for high-resolution applications where $P$ is large.

### **Revisions/Additions during rebuttal**
are visible in **blue ink the updated manuscript**:
- **ImageNet-1K** (Table 1): We added ImageNet-1K results for each setting.
- **Additional experiments with Masked Autoencoder (MAE)** (new Appx. A.6): We evaluated MAEs (trained via patch reconstruction, without CLS tokens). While AAT performs best, our Multi-Layer fusion still is second-best and achieves $22.4-24.7$ [pp] gains over last-layer probes, demonstrating that spatial and hierarchical aggregation are orthogonal.
- **Comparing per-layer linear and attentive probe performance** (new Appx. A.8): Per-layer analysis confirms intermediate layers hold complementary information and encode rich spatial information across patch tokens that summary tokens alone cannot fully capture. A hybrid probe combining all tokens from layers {3, 6, 9} with the final layer demonstrates **orthogonality of hierarchical and spatial aggregation, achieving 98% accuracy on GTSRB**—versus 88.5% for AAT and 84.9% for our method.
- **Finetuning comparison** (new Appx. A.13): While Finetuning is more powerful, our method approaches its performance with 36 times lower training time.
- **Multi-layer fusion across attention probe architectures** (new Appx. A.14): Four alternative attention architectures all benefit from multi-layer fusion, confirming robustness to probe design.
- **Refined scope and applicability** (updated Introduction and Discussion): We clarified that Multi-Layer Fusion is tailored to CLS-token models and tasks driven by global, not patch-level, semantic cues.

**Conclusion**: Our work introduces a **simple adaptive approach to use complementary task-relevant information hidden in intermediate layers**. Hierarchical fusion consistently improves over standard probing (20/20 datasets) while providing interpretable insights into which layers matter for different domains. Offering greater stability than AAT, our method scales efficiently with depth (not resolution), ensuring robustness across diverse and unknown downstream tasks. We further demonstrate that the orthogonality between hierarchical and spatial aggregation opens a principled, powerful path for optimizing information fusion.

---

### Meta-Review · Area_Chair_YEmS · 2025-12-20

**Summary:**

This paper looks at probing-based adaptation for Vision Transformers and introduces a task-dependent attentive probing method. The main idea is that intermediate layers already contain useful information for downstream tasks, as shown in prior work. Instead of relying only on the final layer, the method aggregates representative tokens such as CLS tokens and averaged token embeddings from multiple layers. By attentively combining these intermediate representations, the approach is able to capture complementary information and achieve better performance than standard linear probing across several tasks.

**Reviewer Concerns:**

Several concerns remain.

**First, the gains over the strongest attentive baseline are small and inconsistent.** The core claim is that attending over all layers is better than attending only over the last layer, but the reported improvements are incremental only. For CLIP models the average gain is about one percentage point, for DINOv2 it can slightly decrease, and for supervised ViTs it is roughly comparable. This suggests the method is more of a refinement than a clearly superior alternative, so the paper appears to overstate its impact.

**Second, the practical value of the method is unclear.** While the proposed approach consistently outperforms linear probing, its performance is largely comparable to attentive probing over last-layer tokens. Beyond slightly improved stability, it does not demonstrate a clear or consistent advantage. As a result, the paper does not clearly articulate when or why a practitioner should prefer this more complex multi-layer probing strategy over the simpler and well-established practice of probing only the final layer.

**Third, the choice of baselines is limited.** The paper mainly compares against linear probing, which is better viewed as an evaluation protocol rather than a competitive transfer learning method. In a transfer learning setting, stronger baselines such as full fine-tuning or parameter-efficient approaches like LoRA would provide a more meaningful comparison.

**Reviewer Scores:**

The initial scores were 6, 2, 4, and 4.

Reviewer MGWs, who gave a score of 6 accompanied by very brief comments, is likely to maintain the positive assessment.

Reviewer 5xNU, who gave a score of 2, did not participate in the discussion. While their score might increase, it would likely rise to at most 4, since their main concern about the inappropriateness of the chosen baselines was not fully addressed.

Reviewer XzCg, who initially gave a score of 4, raised their score to 6 after the rebuttal, while noting that the reported gains remain incremental.

Reviewer FHUv, who also gave a score of 4, engaged in the rebuttal but ultimately chose to maintain their original evaluation.

---

### Decision · Program_Chairs · 2026-01-26

Reject